# Consistent Estimation of Identifiable Nonparametric Mixture Models from Grouped Observations

**Alexander Ritchie**[*]
Department of EECS
University of Michigan
Ann Arbor, MI 48109
aritch@umich.edu

**Robert A. Vandermeulen**[*]
ML group
Technische Universität Berlin
10587 Berlin, Germany
vandermeulen@tu-berlin.de

**Clayton Scott**
Departments of EECS, Statistics
University of Michigan
Ann Arbor, MI 48109
clayscot@umich.edu

## Abstract

Recent research has established sufficient conditions for finite mixture models to be identifiable from grouped observations. These conditions allow the mixture components to be nonparametric and have substantial (or even total) overlap. This work proposes an algorithm that consistently estimates any identifiable mixture model from grouped observations. Our analysis leverages an oracle inequality for weighted kernel density estimators of the distribution on groups, together with a general result showing that consistent estimation of the distribution on groups implies consistent estimation of mixture components. A practical implementation is provided for paired observations, and the approach is shown to outperform existing methods, especially when mixture components overlap significantly.

## 1 Introduction

In statistics and machine learning, finite mixture models are often used to describe the distribution of subpopulations within a larger population. A finite mixture model can be written

$$p = \sum_{m=1}^{M} w_m^* p_m^*, \tag{1}$$

where $w_m^* > 0$ are mixing weights such that $\sum_{m=1}^{M} w_m^* = 1$, and $p_m^*$ are probability densities. Without additional assumptions, the mixture model $p$ is not identifiable from iid data. Typically, identifiability is ensured by restricting the $p_m^*$ to some family of parametric distributions. Restricting the $p_m^*$ to be Gaussian yields the Gaussian mixture model (GMM) which is identifiable [1, 2].

Most work on estimating mixture models assumes an iid sampling scheme. In this work we examine an alternative sampling scheme where observations occur in iid groups. Each group is generated by sampling a component $m \in [M]$ according to $w_m^*$, and then drawing $N$ iid observations from $p_m^*$.

Recent work has shown that *any* finite mixture model is identifiable given grouped observations of sufficient size [3]. In the worst case, any finite mixture model with $M$ components is identifiable given groups of size $N \geq 2M - 1$. It was also shown that, if the underlying components of the mixture model are *jointly irreducible* [4], then the mixture is identifiable given paired observations ($N = 2$). This framework provides a setting where the potential exists to recover nonparametric and highly overlapping mixture components. As of yet, however, no general theory or algorithms are known for this estimation problem.

This work makes the following contributions. We introduce a novel variant of the kernel density estimator that yields statistically consistent estimates of *any identifiable* nonparametric mixture

---

[*]Equal contribution.

model (NoMM) from grouped observations. To prove this result, we establish an oracle inequality for weighted kernel density estimators. We also establish a general result showing that consistent estimation (with an estimator possessing a natural factored form) of the distribution on groups implies consistent estimates of the underlying components when the NoMM is identifiable. The only additional condition imposed by our theory is that the $p_m^*$ be square integrable. In the case of $N = 2$, we offer an efficient algorithm and demonstrate its effectiveness on several datasets.

We study two applications where paired observations naturally arise. The first is nuclear source detection, where nuclear particles interact with a detector to produce some form of measurement. A critical challenge in this application is to classify incoming particles as belonging to source or background. Because of changing environments, training data are typically not available, and these two classes also have substantial overlap. By positioning two detectors side-by-side, it is possible to simultaneously measure two particles from the same (unknown) class.

We also apply our method to topic modeling of Twitter data. Since tweets usually express a small set of very closely related ideas, words in tweets contain common underlying semantic information. The pairing of words has the potential to encode this semantic information in a way that accounts for context. The proposed method, which operates on continuous word embeddings, allows for flexible modeling of the distributions of topics over words using static word embeddings [5]. Furthermore our method does not require anchor words, allows for substantial overlap of topics without loss of identifiability, and can be trained using documents with as few as two words *without* any document aggregation [6, 7]. While other works have explored topic modeling with word embeddings, which we call continuous topic modeling, most either impose parametric assumptions or are not suited for very short texts. To our knowledge, this is the first work to consider a nonparametric approach to continuous topic modeling of very short texts.

## 2    Background and Previous Work

Much of the literature concerning NoMMs falls in the category of Bayesian nonparametrics, a thorough summary of which is given in [8]. Typically, mixture models in this setting do not assume that the number of mixture components is known, and instead assume that the mixture components are from a known parametric family of distributions. An in-depth treatment of Bayesian NoMMs (BNoMMs) can be found in [9]. The parametric assumptions on the mixture components have been relaxed in [10], but the identifiability results impose regularity and separation conditions on the components. We mention BNoMMs only for completeness and emphasize that our work considers an alternative setting where the number of mixture components is known, but few to no assumptions are made on the mixture components themselves.

Mixture models are often utilized to solve the clustering problem. Parametric mixture models, such as GMMs, are able to capture overlapping clusters. Most clustering algorithms, however, such as $k$-means [11, 12], DBSCAN [13], and spectral clustering [14, 15], assume clusters are non-overlapping and hence fail when clusters overlap. The grouped observation setting considered in this work is known in the clustering literature as *clustering with instance-level constraints* [16, 17, 18]. A survey of constrained clustering is given in [19]. Grouped observations correspond to so-called must-link constraints, where two or more observations are known, through expert knowledge or some other means, to belong to the same cluster. Most constrained clustering approaches cannot model overlapping clusters effectively [20].

There is relatively little work on mixture modeling with nonparametric components, and to our knowledge no prior work addresses the incorporation of instance-level constraints in the NoMM setting. Mallapragada et al. [21] use a mixture of kernel density estimators to estimate a NoMM, but do not address identifiability or provide statistical guarantees. Aragam et al. [10] prove identifiability of NoMMs under regularity and separation conditions on the components. Schiebinger et al. [22] study kernelized spectral clustering and characterize recoverability of components with small overlap. Zheng and Wu [23] establish consistent estimation of NoMMs under the assumption that mixture components have independent marginals. Bao et al. [24] consider the related problem of "similar-unsupervised" binary classification, which assumes access to unlabeled data in addition to must-link constraints.

In the grouped observation setting, previous works on multi-view models can be adapted to prove identifiability results and give algorithms to recover mixture model components. When the mixture

components are linearly independent it has been shown that three observations per group is sufficient to yield identifiability as well as an algorithm to provably recover the components [25, 26]. We note that these approaches require three observations per group, while the proposed method works with as few as two observations per group. This difference amounts to performing kernel density estimation in three times the ambient dimension versus two. With the instability of KDEs in high dimension, the reduction to groups of size two can be very meaningful in practice. Furthermore, in applications like nuclear particle classification, triples may be exceedingly rare or difficult to measure. For discrete data, similar results from nonnegative matrix factorization exist under joint irreducibility with two observations per group [27], and algorithms have been proposed to recover arbitrary mixture models with $M$ components given $2M - 1$ observations per group [28, 3].

## 3  Problem Statement

**Notation**   For $1 \leq p < \infty$ denote $\|f\|_p := (\int_{\mathbb{R}^d} |f(x)|^p dx)^{1/p}$, and $L^p := \{f : \mathbb{R}^d \to \mathbb{R} : \|f\|_p < \infty\}$. The transpose of a matrix $A$ will be written $A'$. Random variables will be referred to by capital letters, and instances of random variables will be referred to by the corresponding lowercase letter. We represent the set of positive integers $\{1, 2, \ldots, M\}$ by $[M]$. We let $\Delta^R$ be the probability simplex in $\mathbb{R}^R$. We denote the $M$-fold Cartesian product of a set with a subscript, e.g., $\Delta^R_M = \underbrace{\Delta^R \times \ldots \times \Delta^R}_{M}$.

We precisely introduce the grouped observation setting, review known identifiability results, and formalize the estimation problem. We assume $M$, the number of mixture components, is known. While this may seem like a strong assumption, it runs counter to the assumptions used in BNoMMs, where $M$ is modeled nonparametrically but mixture components are assumed to be parametric distributions. In practice, $M$ can be estimated by looking for the knee in the scree plot of the initialization we suggest. Alternatively, an approach based on the Bayesian information criterion or minimum description length could be useful, but we leave this to future work.

The standard sampling procedure for a mixture model of the form $p = \sum_{m=1}^{M} w_m^* p_m^*$ can be viewed as a two step process wherein one samples a mixture component $p_m^*$ with probability $w_m^*$ and then observes one draw from that distribution $X \sim p_m^*$. The *grouped observation setting* considers an alternative sampling scheme where, after selecting a mixture component $p_m^*$, instead of only drawing a single observation, a group of observations $\mathbf{X} = (X_1, \ldots, X_N)$ are drawn iid from $p_m^*$. As in a standard mixture model, one does not know *a priori* from which mixture component a grouped observation is sampled. Repeating this $n$ times, one's data consists of $n$ groups of $N$ observations per group $\mathbf{X}_1 = (X_{1,1}, \ldots, X_{1,N}), \ldots, \mathbf{X}_n = (X_{n,1}, \ldots, X_{n,N})$. The distribution on groups is $\mathbf{X} \overset{iid}{\sim} \sum_{m=1}^{M} w_m^* p_m^{* \times N}$, where $p_m^{* \times N} : \mathbb{R}^d_N \to \mathbb{R}$ denotes the product density such that $p_m^{* \times N}(y_1, y_2, \ldots, y_N) = p_m^*(y_1) p_m^*(y_2) \ldots p_m^*(y_N)$. Note that when $N = 1$ this is simply a standard mixture model.

Vandermeulen and Scott [3] characterized identifiability from grouped observations for mixtures of general probability measures. A mixture model $p = \sum_{m=1}^{M} w_m^* p_m^*$ is said to be *N-identifiable* if $p$ cannot be expressed $p = \sum_{m=1}^{M'} w_m' p_m'$ for some distinct mixture model such $M' \leq M$ and $\sum_{m=1}^{M} w_m^* p_m^{* \times N} = \sum_{m=1}^{M'} w_m' p_m'^{\times N}$. In words, $N$-identifiability of $p$ means there is no other mixture model with $M$ or fewer components that induces the same distribution on groups. They show that a general mixture model is $N$-identifiable from grouped observations provided $N \geq 2M - 1$, and that this cannot be improved without imposing restrictions on the components. The result places no assumptions whatsoever on the components.

In practice, the bound of $2M - 1$ is probably pessimistic, and the most useful cases are likely when $N$ is small, say two or three. The authors of [3] also show that if the $p_m^*$ are *jointly irreducible* (*linearly independent*), then the mixture is $N$-identifiable for $N = 2$ ($N = 3$). A collection of probability densities $\mu_1, \mu_2, \ldots, \mu_M$ is said to be *jointly irreducible* (JI) if $\sum_{m=1}^{M} c_m \mu_m$ is never a valid density whenever some $c_m < 0$. JI is satisfied, for example, if the support of each mixture component has some subset of positive measure that does not intersect the supports of the other mixture components (a continuous analogue of the anchor word assumption). This is not necessary, however; JI is still possible if all densities have the same support. In the remainder of the paper we focus on the setting of $N = 2$, not only because JI provides a flexible nonparametric condition where paired observations

suffice, but also because the notation for our estimator becomes cumbersome when $N > 2$. Our theory generalizes easily to $N > 2$, and these details our described in the supplemental material.

The paired observations $\mathbf{X}_1, \ldots, \mathbf{X}_n$ with $\mathbf{X}_i = (X_{i,1}, X_{i,2}) \in \mathbb{R}^d \times \mathbb{R}^d$ are iid and have density

$$q(x, x') := \sum_{m=1}^{M} w_m^* p_m^*(x) p_m^*(x') \quad x, x' \in \mathbb{R}^d.$$

We assume $M$ is known. *Our goal is to consistently estimate $w_m^*$ and $p_m^*$ when $p$ is identifiable.*

We now briefly motivate the grouped observation setting. Drawing a comparison to existing work, our proposed method can be considered a continuous version of multinomial mixture modeling, which is used in psychometrics where measurements over time are collected for a group of, for example, bipolar disorder patients and used to identify subgroups within that population whose condition is only evident with repeated temporal measurements [29]. Additionally, many mixture modeling problems can be transformed to the grouped observation setting by adjusting the sampling procedure, making simplifying assumptions on existing data, or by manual grouping based on domain expert knowledge. For example, in the Twitter experiment we assume tweets usually have only a single topic, suggesting grouped observations can be created by selecting words at random from a given tweet. In a sensor network, assuming stationarity of the measured process, one could double sensors in each location or sample twice in quick succession, rather than once, at the sampling interval. Much as iid assumptions on non-grouped observations are used to simplify analysis and hold only approximately in collected data, the same can be said about collecting grouped observations.

## 4 A Weighted Kernel Density Estimator

Our overall strategy is to first devise a consistent estimator of $q$, the density on pairs, where the estimator has a factorized form reflecting the group sampling scheme. In the next section we prove that if an estimator for $q$ is consistent, and $p$ is identifiable, then the components comprising our estimator converge to the true components.

Let $k : \mathbb{R}^d \to \mathbb{R}$ be a function, called a *kernel*, such that $k \geq 0$ and $\int k(x)dx = 1$. An example is the Gaussian kernel $k(x) = (2\pi)^{-1/2} \exp(-\|x\|^2/2)$. For $\sigma > 0$, define $k_\sigma(x, x') := \sigma^{-d} k((x-x')/\sigma)$. We refer to the second argument of $k_\sigma$ as the *center* of the kernel. A *weighted kernel density estimator* (wKDE) for a density on $\mathbb{R}^d$, but constructed from the paired observations $\mathbf{X}_i$, has the form

$$p(x; \theta) = \sum_{r=1}^{n} \sum_{r'=1}^{2} \theta_{r,r'} k_\sigma(x, X_{r,r'}),$$

where $\theta_{r,r'}$ is the element of $\theta = [\theta_{1,1}, \theta_{1,2}, \ldots, \theta_{n,1}, \theta_{n,2}]'$ corresponding to the weight of the kernel centered at $X_{r,r'}$. We propose to model the mixture components as wKDEs. Specifically, given $n$ paired observations, we consider estimators of $q$ of the form

$$q_{w,\alpha}(x, x') = \sum_{m=1}^{M} w_m p(x; \alpha_m) p(x'; \alpha_m), \tag{2}$$

where $w = [w_1, w_2, \ldots, w_M]' \in \Delta^M$, $\alpha_m = [\alpha_{m,1,1}, \alpha_{m,1,2}, \ldots, \alpha_{m,n,1}, \alpha_{m,n,2}]' \in \Delta^{2n}$ for all $m \in [M]$, with $\alpha_{m,r,r'}$ corresponding to the weight of the kernel centered at $X_{r,r'}$ in the estimate of the $m^{th}$ mixture component, and $\alpha := (\alpha_1, \alpha_2, \ldots, \alpha_M) \in \Delta_M^{2n}$.

To select the parameters $(w, \alpha)$, we propose to minimize the integrated square error (ISE) of $q_{w,\alpha}$ given by $\|q - q_{w,a}\|_2^2 := \int [q(x, x') - q_{w,a}(x, x')]^2 dx dx'$. Expanding the ISE gives

$$\|q - q_{w,a}\|_2^2 = \int q_{w,\alpha}^2(x, x') dx dx' - 2 \int q_{w,\alpha}(x, x') q(x, x') dx dx' + \overbrace{\int q^2(x, x') dx dx'}^{\text{const.}} .$$

Since the final term is constant with respect to $w$ and $\alpha$, we focus on minimizing the first two terms which we call the truncated ISE (TISE) and denote by $J(w, \alpha)$. Substituting the definition of $q_{w,\alpha}$ in the TISE yields

$$J(w, \alpha) := \int q_{w,\alpha}^2(x, x') dx dx' - 2 \sum_{m=1}^{M} \sum_{r=1}^{n} \sum_{r'=1}^{2} \sum_{s=1}^{n} \sum_{s'=1}^{2} w_m \alpha_{m,r,r'} \alpha_{m,s,s'} h(r, r', s, s'), \tag{3}$$

where $h(r, r', s, s') \coloneqq \int k_\sigma(x, X_{r,r'}) k_\sigma(x', X_{s,s'}) q(x, x') dx dx'$. Since $q$ is unknown, the ISE and therefore $J(w, a)$ cannot be calculated directly. Noting that $h(r, r', s, s')$ is an expectation, we estimate this term using a hybrid leave-one-out/leave-two-out (LOO/LTO) estimator

$$\hat{h}(r, r', s, s') \coloneqq \begin{cases} \frac{1}{n-2} \sum_{i \in [n] \setminus \{r,s\}} k_\sigma(X_{i,1}, X_{r,r'}) k_\sigma(X_{i,2}, X_{s,s'}) & r \neq s \\ \frac{1}{n-1} \sum_{i \in [n] \setminus \{r\}} k_\sigma(X_{i,1}, X_{r,r'}) k_\sigma(X_{i,2}, X_{s,s'}) & r = s \end{cases}.$$

In this manner we have the empirical TISE

$$\hat{J}(w, \alpha) \coloneqq \int q_{w,\alpha}^2(x, x') dx dx' - 2 \sum_{m=1}^{M} \sum_{r=1}^{n} \sum_{r'=1}^{2} \sum_{s=1}^{n} \sum_{s'=1}^{2} w_m \alpha_{m,r,r'} \alpha_{m,s,s'} \hat{h}(r, r', s, s'). \quad (4)$$

With all the notation in place, our estimate of the nonparametric mixture model is determined by

$$(\hat{w}, \hat{\alpha}) \coloneqq \underset{w \in \Delta^M, \ \alpha \in \Delta_M^{2n}}{\arg\min} \hat{J}(w, \alpha), \quad (5)$$

where $\hat{w}_m$ are the mixing weights and $p(x; \hat{\alpha}_m)$ are the mixture components for $m \in [M]$. The theoretical results presented in Section 5 concern the behavior of the minimizer of (5). We show not only that the empirical TISE minimizing estimator $\hat{q} \coloneqq q_{\hat{w}, \hat{\alpha}}$ consistently estimates $q$, but its components also consistently estimate the underlying mixture model if it is identifiable.

## 5 Theoretical Results

In this section we state our assumptions and main results. Formal proofs are given in the supplemental material. Our overall approach is to first show that the proposed $\hat{q}$ is a consistent estimate of $q$ (Theorems 1 and 2). We then show that if $p$ in (1) is identifiable, then the components $p(x; \hat{\alpha}_m)$ defining $\hat{q}$ are consistent estimates of $p_m^*$, as are the $\hat{w}_m$ for $w_m^*$ (Theorem 3).

We assume throughout this section that $p_m^* \in L^2$ for all $m$. We also require that the kernel $k$ satisfy two additional conditions: $k \in L^2$ and $k \leq C_k$ for some constant $C_k < \infty$.

We begin with an oracle inequality, which shows that our estimator selects an approximately optimal member of our model class.

**Theorem 1.** *Let $\epsilon > 0$ and set $\delta = 8(n^2 - n) \exp\{-\frac{\sigma^{4d}(n-2)\epsilon^2}{8C_k^4}\} + 8n \exp\{-\frac{\sigma^{4d}(n-1)\epsilon^2}{8C_k^4}\}$. With probability at least $1 - \delta$ the following holds: $\|q - q_{\hat{w},\hat{\alpha}}\|_2^2 \leq \inf_{w \in \Delta^M, \ \alpha \in \Delta_M^{2n}} \|q - q_{w,\alpha}\|_2^2 + \epsilon$.*

*Proof Sketch.* The estimators $\hat{h}$ are constructed so that they are sums of independent random variables, allowing us to apply Hoeffding's inequality to show that each $\hat{h}$ concentrates around its $h$. Then using basic inequalities (triangle inequality, union bound) and the simplex constraints on $w$ and $\alpha$, we show that $\hat{J}(w, \alpha)$ concentrates around $J(w, \alpha)$ uniformly over the parameter space. $\quad\square$

The next result uses Theorem 1 to establish that $\hat{q}$ is a consistent estimate of $q$ in the $L^1$ norm.

**Theorem 2.** *If $\sigma \to 0$ and $\frac{n\sigma^{4d}}{\log n} \to \infty$ as $n \to \infty$, then $\|q - q_{\hat{w},\hat{\alpha}}\|_1 \xrightarrow{a.s.} 0$.*

*Proof Sketch.* We appeal to a result of [30] showing that if $\int \hat{q} = 1$, which it does in our case, then strong consistency (i.e., a.s. convergence) of a density estimator in $L^2$ implies strong consistency in $L^1$. To show strong consistency in $L^2$, from Theorem 1 it suffices to exhibit $w \in \Delta^M$ and $\alpha \in \Delta_M^{2n}$ such that $\|q - q_{w,\alpha}\|_1 \xrightarrow{a.s.} 0$. For this we take $w = w^*$ and $\alpha = \alpha^*$ such that each $\alpha_m^*$ is uniform on the data points drawn from $p_m^*$. This makes $p(\cdot; \alpha_m^*)$ the usual (uniformly weighted) KDE for $p_m^*$, which is known to be a strongly consistent estimator. Strong consistency of $\hat{q}$ then easily follows. $\quad\square$

The preceding results hold regardless of whether $p$ in (1) is identifiable. The next result states that if $p$ is identifiable, then the estimates $p(\cdot; \hat{\alpha}_m)$ comprising $\hat{q}$ are consistent estimates of the true components $p_m^*$, as are the $\hat{w}_m$ of $w_m^*$. The result is stated for $N \geq 2$.

**Theorem 3.** *Let $\sum_{m=1}^{M} w_m p_m$ be an $N$-identifiable mixture model, and $\sum_{m=1}^{M} \hat{w}_{m,j}\hat{p}_{m,j}$ be a sequence of mixture models such that $\left\|\sum_{m=1}^{M} \hat{w}_{m,j}\hat{p}_{m,j}^{\times N} - \sum_{m=1}^{M} w_m p_m^{\times N}\right\|_1 \to 0$. Then there is a sequence of permutations $\sigma_j$ so that $\hat{w}_{\sigma_j(m),j} \to w_m$ and $\left\|\hat{p}_{\sigma_j(m),j} - p_m\right\|_1 \to 0$ for all $m$.*

*Proof Sketch.* We show that if $\left\|\sum_{m=1}^{M} \hat{w}_{m,j}\hat{p}_{m,j}^{\times N} - \sum_{m=1}^{M} w_m p_m^{\times N}\right\|_1 \to 0$ then the components $\hat{p}_{m,j}$ admit some convergent subsequence, and therefore so do $\hat{p}_{m,j}^{\times N}$. If a subsequence $\hat{p}_{m,j}^{\times N}$ stays away from the components $p_m^{\times N}$ then some subsequence would converge to a component other than some $p_m^{\times N}$. This allows us to construct a mixture model violating $N$-identifiability, a contradiction. $\quad\square$

This result has been stated in terms of densities for readability, but the supplemental contains a general measure-theoretic version. We may combine Theorems 2 and 3 to establish the following (returning to the setting of $N = 2$). To our knowledge, this is the first result to establish consistent estimation, under any sampling scheme, of NoMMs with substantial overlap.

**Corollary 1.** *If $\sigma \to 0$ and $\frac{n\sigma^{4d}}{\log n} \to \infty$ as $n \to \infty$, and $p$ is 2-identifiable (e.g., the $p_m^*$ are jointly irreducible), then $\hat{w}_m \overset{a.s.}{\to} w_m^*$ and $\|p(\cdot\,; \hat{\alpha}_m) - p_m^*\|_1 \overset{a.s.}{\to} 0$, up to a permutation.*

The significance of the result is that joint irreducibility is both a flexible nonparametric assumption, while ensuring identifiability in the case $N = 2$ for which a practical implementation of $\hat{q}$ is possible. We include an analogous result for *all identifiable NoMMs* in the supplement. Finally, we mention that recent results on nonparametric estimation of densities which are convex combinations of separable densities [31] suggest that it may be possible to remove the 4 in the $\sigma^{4d}$ term in our rates.

## 6 Optimization

In this section we suggest an approach for solving (5). We first consider the problem as presented up to this point, which we call the *full problem*. We then consider an approach for speeding up optimization by heuristically choosing a coreset as the kernel centers, which we call the *coreset approach*. In what follows, we assume that $\widetilde{k}_\sigma(z_r, z_u) := \int k_\sigma(x, z_r)k_\sigma(x, z_u)dx$ has a closed-form expression or can otherwise be computed efficiently. This assumption is satisfied by many common kernels such as the Gaussian, Cauchy, and Laplacian kernels.

**Form of the Optimization Problem.** The optimization problem (5) can be written

$$\min_{w \in \Delta^M,\ \alpha \in \Delta_M^R} \sum_{k=1}^{M}\sum_{\ell=1}^{M} w_k w_\ell \left(\alpha_k' G \alpha_\ell\right)^2 - 2\sum_{m=1}^{M} w_m \left(\alpha_m' C \alpha_m\right), \tag{6}$$

where the matrices $G, C \in \mathbb{R}^{R \times R}$ will be defined shortly. Details are given in section S2.1 of the supplementary material. In particular, both the full problem and the coreset approach can be written in the form of (6), differing only in the definitions of $R$ and $G, C$. We therefore propose to use the same optimization approach for both problems. For the full problem, $R = 2n$ and the matrices $G$ and $C$ have the form

$$G_{a,b} = \widetilde{k}_\sigma\left(X_{\lfloor \frac{a}{2}\rfloor, a \bmod 2}, X_{\lfloor \frac{b}{2}\rfloor, b \bmod 2}\right) \quad C_{a,b} = \hat{h}\left(\left\lfloor \frac{a}{2}\right\rfloor, a \bmod 2, \left\lfloor \frac{b}{2}\right\rfloor, b \bmod 2\right).$$

Though the problem (6) is nonconvex, we observe that a properly initialized alternating projected stochastic gradient descent (APSGD) procedure produces good solutions in practice. We do not make any claims about the convergence of our algorithm. We do however note that the objective is a degree six polynomial and that the constraint set is convex. Projected SGD is commonly used in practice to solve large-scale constrained nonconvex maachine learning problems, and convergence to a stationary point has been explored in the literature for various settings [32].

Pseudocode for the APSGD algorithm for solving (6) is given in the supplementary material. We mention that the projections are onto the probability simplex, a decaying step size is used, and stochasticity is introduced via the matrix $C^{(t)}$, which is a mini-batch version of $C$ defined by $C_{a,b}^{(t)} = \frac{1}{|\Omega^{(t)}\setminus\{a,b\}|}\sum_{i \in |\Omega^{(t)}\setminus\{a,b\}|} k_\sigma(X_{i,1}, X_{\lfloor \frac{a}{2}\rfloor, a \bmod 2})k_\sigma(X_{i,2}, X_{\lfloor \frac{b}{2}\rfloor, b \bmod 2})$, where $\Omega^{(t)}$ is the index set corresponding to the $t^{th}$ mini-batch.

**Coreset Approach.** KDEs traditionally center kernels at the location of each observation, i.e., $k_\sigma(\cdot, x_{i,i'})$, where $x_{i,i'}$ is the kernel center. Rather than constraining the wKDE to have kernels centered at the observations, we can formulate the optimization problem with $R$ kernel centers $z_r \in \mathbb{R}^d$ for some suitably chosen $z_r$, which we take to be our coreset. Further details are given in the supplementary material. We note the per-batch computational complexity for our APSGD algorithm is dominated by the gradient calculations and calculating $C^{(t)}$. If we assume $R > M$, the total complexity is $\mathcal{O}(n_e n(M + d)R^2)$ where $n_e$ is the number of training epochs. Thus, choosing $R \ll 2n$ offers a substantial speed-up.

**Initialization.** We adopt a spectral initialization scheme. We focus on the full problem for concision, but the coreset approach is similar; further details for both are provided in the supplementary material. By Lemmas 5.1 and 8.2 of Vandermeulen and Scott [3], one can view the standard KDE on the full sample as a symmetric linear operator $T : L^2(\mathbb{R}^d) \to L^2(\mathbb{R}^d)$. We use the eigenvectors of $T$, which are wKDEs on $\mathbb{R}^d$, to form a low-rank approximation of the standard KDE initialize our algorithm. This initialization is a low-rank approximation of the standard KDE. Further details for both the full problem and the coreset approach are provided in the supplementary material.

# 7 Experiments

In this section we compare our coreset approach against several competing methods on a number of real and highly overlapping synthetic datasets. Datasets are described in Table 1. We call the proposed method Nonparametric Density estimation of Identifiable mixture models from Grouped Observations (NDIGO). All code and synthetic datasets are publicly available.[2] The MAGIC gamma ray detection dataset [33] is publicly available via the UCI machine learning repository. The Russian-troll-tweets Twitter dataset is publicly available through FiveThirtyEight.[3] For NDIGO and MVLVM, we used a Gaussian kernel in all experiments and Scott's rule [34] was used for bandwidth selection. For synthetic experiments, $R$ was selected to yield the initialization with the lowest empirical TISE. $R$ was chosen from $\{10, 20, 30, 40, 50\}$ for both moons datasets, and from $\{60, 70, 80, 90, 100\}$ for the Olympic rings and half-disks datasets. We used $R = 200$ for the MAGIC and Twitter datasets.

Several of the methods we compare against do not produce density estimates, so we evaluate the clustering induced by each method. For constrained clustering methods, we compare against constrained spectral clustering (CSC) [35], and constrained GMM (CGMM) [36]. We also compare against the NoMM methods NPMIX of Aragam et al. [10] and MVLVM of Song et al. [37]. MVLVM is our most similar competitor as it considers groups of size three. Each constrained clustering algorithm was given access to all pair information. MVLVM was supplied triplets from the training data. NPMIX does not utilize the pair information in any way. Following the literature, we report the clustering results for the training sample. Out-of-sample results are provided in the supplementary material, but we mention NDIGO is the best performer. Parameters for CSC and NPMIX were optimized w.r.t. a separately generated holdout dataset. Average results over ten runs on the synthetic datasets are shown in Figure 1. NDIGO outperforms all methods considered. The synthetic datasets were constructed to have clusters that are non-ellipsoidal in shape with substantial overlap between clusters. The clusterings induced by each method are shown in Figure 1. Performance is measured in terms of the adjusted Rand index (ARI) [38]. We observe that NDIGO gives superior performance across all experiments, especially when clusters have substantial overlap. Density estimates produced by our method for synthetic datasets are shown in Figure 2.

Results on the MAGIC dataset are shown in Figure 3. The task is to detect gamma radiation events among background radiation. When detecting rare events, the proper performance indicator is given by the receiver operating characteristic (ROC) curve, which plots the true positive rate vs. the false positive rate, parameterized by the threshold of a likelihood ratio test (LRT). Each method was trained using $80\%$ of the available data, and the ROC curve was generated from the remaining $20\%$. CSC was excluded from this test because it does not produce a density estimate, so a LRT cannot be applied. As an upper bound on possible unsupervised performance, we trained KDEs on each class and plugged the resulting density estimates into an LRT. Previous studies concluded this method, which we call KDE-plugin, is the best approach [33]. We find NDIGO and CGMM perform very similarly in this experiment, outperforming other methods and approaching KDE-plugin.

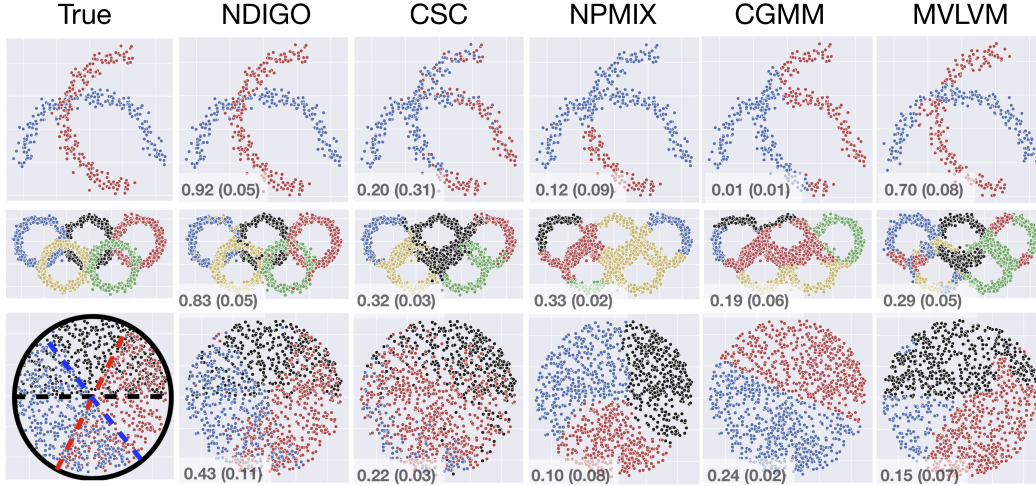

Figure 1: Example cluster assignments of three synthetic datasets by each method. Mean ARI (standard deviation) over 10 runs is shown at the bottom left of each clustering (larger is better).The datasets are overlapping moons (top), Olympic rings (middle), and half-disks (bottom). Half-disks has been annotated to show the true components.

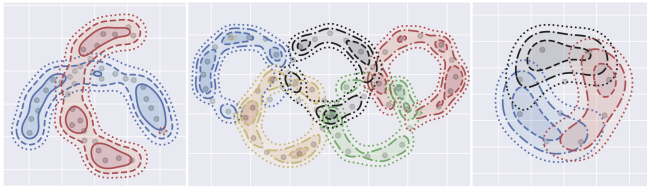

Figure 2: Component density estimate contours produced NDIGO. From left to right: overlapping moons, Olympic rings, half-disks

Table 1: Description of datasets. *Quantities after preprocessing.

| Dataset (2n) | $M/d$ |
|---|---|
| Ovlp. Moons $(400)$ | 2/2 |
| Olympic Rings $(2000)$ | 5/2 |
| Half-disks $(1200)$ | 3/2 |
| MAGIC $(19,020)$ | 2/10 |
| Twitter $(3,382,162^*)$ | -/10* |

We applied NDIGO to topic modeling on the Twitter dataset. Results are shown in Tables 2 and 3. Details of data preprocessing are deferred to the supplementary material. After preprocessing, the dataset consisted of $1,691,081$ pairs of 10-dimensional embedded words where each element in a pair comes from the same tweet. Algorithms for competing methods, as described by their respective authors, could not scale to this experiment. Therefore, we compare to recent methods designed for continuous topic modeling of short texts: LF-DMM [39], and GPU-DMM[40] as implemented by

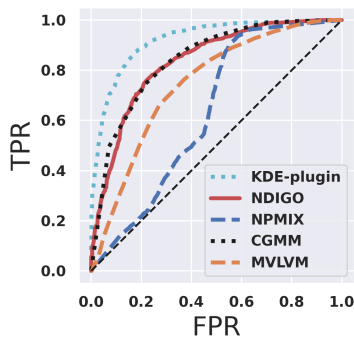

Figure 3: Receiver operating characteristic for MAGIC gamma ray detection dataset.

Table 2: Russian-troll talking points learned from Twitter dataset. * Censored (racial epithet)

| Topic | Selected Top 10 Words |
|---|---|
| 1 | dead, man, kill, missing, families, young |
| 2 | make, good, better, enough, yet, even, get |
| 3 | politics, inside, news, local, police, new, state |
| 4 | trial, a*, gentrified, wk, deport, b* |
| 5 | businesses, competitive, strength, people, white |

Table 3: Mean and standard deviation of topic coherence on Twitter dataset over five experiments.

| NDIGO | LF-DMM | GPU-DMM |
|---|---|---|
| $0.521 \pm 0.084$ | $0.493 \pm 0.018$ | $0.435 \pm 0.009$ |

Qiang et al. [41].[4] A selection of the top 10 words of topics uncovered by NDIGO is given in Table 2. We find that the discovered topics correspond well to other analyses of the dataset [42]. Using topic coherence (pointwise mutual information) as an evaluation metric [43], we observe that NDIGO is competitive with the competing methods.

# 8 Conclusion

In this work we introduced a novel variant of the kernel density estimator that yields consistent estimates of any identifiable nonparametric mixture model from grouped observations. We established an oracle inequality for weighted kernel density estimators, and a general consistency result for estimators of the form $q_{w,\alpha}$. Namely, consistent estimation of $q$ implies consistent estimates of the underlying components when the NoMM is identifiable. In the case of $N = 2$, we offer an efficient algorithm and demonstrate its effectiveness on several datasets where traditional approaches fail. Additionally, we show our approach has practical applications in topic modeling with very small documents and nuclear source detection.

## Broader Impact

Our work could be applied to many problems for which labeling data is prohibitive, but groups of similar data points can be collected easily. We have explored two such applications: nuclear source detection and topic modeling. These applications, as we have interpreted them, have potential for positive societal impact. Since our work is mostly theoretical and centered around a relatively unexplored sampling scheme, there are likely applications we have not anticipated. We note that topic modeling, in general, has potential surveillance applications, but this is not unique to the proposed method.

**Funding in direct support of this work:** AR and CS were supported in part by the National Science Foundation under awards 1838179 and 2008074, by the Department of Defense, Defense Threat Reduction Agency under award HDTRA1-20-2-0002, and by the Michigan Institute for Data Science. RV acknowledges support by the Berlin Institute for the Foundations of Learning and Data (BIFOLD) sponsored by the German Federal Ministry of Education and Research (BMBF).

## Footnotes

[2]Authors' GitHub link to go here in final version.

[3]available online: `https://github.com/fivethirtyeight/russian-troll-tweets`

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
