[Supplementary Material · NoMM_Neurips2020_supplementary.pdf]

# Supplement to Consistent Estimation of Identifiable Nonparametric Mixture Models from Grouped Observations

# Contents

## S1 Additional Experimental Details

In this section we provide details of the Twitter experiment and out-of-sample results for experiments on synthetic datasets.

### S1.1 Out-of-sample Results

Here we provide out-of-sample results for the synthetic experiments shown in the main paper. These are shown in Table 1. We make the realistic assumption that pair information is not available for out-of-sample data. We generate a test dataset $20\%$ the size of the training set according to the distribution of the training data.

Table 1: Out-of-sample ARI and standard deviation over 10 runs on synthetic datasets.

| Dataset | NDIGO | CSC | NPMIX | CGMM | MVLVM |
|---------|-------|-----|-------|------|-------|
| Overlapping moons | $0.705 \pm 0.091$ | $0.405 \pm 0.283$ | $0.131 \pm 0.075$ | $0.002 \pm 0.011$ | $0.593 \pm 0.229$ |
| Olympic Rings | $0.607 \pm 0.058$ | $0.387 \pm 0.033$ | $0.367 \pm 0.063$ | $0.127 \pm 0.012$ | $0.290 \pm 0.053$ |
| Half-disks | $0.221 \pm 0.036$ | $0.215 \pm 0.038$ | $0.102 \pm 0.095$ | $0.185 \pm 0.076$ | $0.127 \pm 0.062$ |

### S1.2 Preprocessing of Twitter Dataset

Twitter dataset is publicly available through FiveThirtyEight.[1] The data consist of tweets, from a variety Russian-troll twitter accounts, tweeted between 2015 and 2018. We considered all tweets from 2016, a total of $878,878$. We pre-processed the tweets by removing stop words, punctuation, and hyperlinks, followed by a lemmatization step and the removal of any words that were not contained in the vocabulary of the six billion token GloVe word vectors [9]. For completeness, we mention that lemmatization is a common pre-processing step in natural language processing that removes inflectional differences from words by mapping each inflection to a common base form called the lemma. For example, lemmatization will map each of the words dog, dogs, dog's, dogs', and doggy to the word dog. For each tweet, we paired the constituent words uniformly at random without replacement, resulting in $1,691,081$ pairs of words where the words of a given pair come from the same tweet. No other information was retained. We emphasize that a given word from a given tweet will *not* be assigned to more than one pair. However, if a given word appears in multiple tweets (which may not all be about the same topic), it will show up in multiple pairs.

For the embedding step we performed PCA on the pre-trained GloVe 50-dimensional embeddings to obtain 10-dimensional vectors which were used to encode the paired words. The kernel centers were obtained by running mini-batch $k$-means with $R = 200$ on a uniform random sample of $10,000$ of the $1,691,081$ word pairs, from which the matrix $G$ was also calculated. We then trained on the full $1,691,081$ word pairs, which are utilized in mini-batches through the matrix $C^{(t)}$.

Algorithms for competing methods, as described by their respective authors, could not scale to this experiment. Therefore, we compare to recent methods designed for continuous topic modeling of short texts: LF-DMM [8], and GPU-DMM[6] as implemented by Qiang et al. [10].[2] LF-DMM and GPU-DMM were trained on the same preprocessed data as NDIGO, where each of the $1,691,081$ word pairs is considered a unique document. Each of these methods were run with default hyperparameters, as described in the documentation for GPU-DMM and LF-DMM.[2] After training, UCI topic coherence [7] (which measures pointwise mutual information) was used to evaluate performance. UCI topic coherence uses a reference dataset to estimate word co-occurrence probabilities, which is more robust in the short text setting as very common words in a given topic may never be observed to co-occur. A recent Wikipedia article dump was used for the reference dataset, and is provided with our code.

## S2 Optimization Details

In this section we provide details of our algorithm, including the form of the objective function and initialization, for both the full problem and the coreset approach.

Recall the expression for the ETISE

$$\hat{J}(w, \alpha) \triangleq \int q^2_{w,\alpha}(x, x')dxdx' - 2 \sum_{m=1}^{M} \sum_{r=1}^{n} \sum_{r'=1}^{2} \sum_{s=1}^{n} \sum_{s'=1}^{2} w_m \alpha_{mrr'} \alpha_{mss'} \hat{h}(r, r', s, s'),$$

where

$$\hat{h}(r, r', s, s') \triangleq \begin{cases} \hat{h}_{\text{LTO}}(r, r', s, s'), & r \neq s \\ \hat{h}_{\text{LOO}}(r, r', s'), & r = s \end{cases}$$

$$\hat{h}_{\text{LOO}}(r, r', r'') \triangleq \frac{1}{n-1} \sum_{i \in [n] \setminus \{r\}} k_\sigma(x_{i,1}, x_{r,r'}) k_\sigma(x_{i,2}, x_{r,r''})$$

$$\hat{h}_{\text{LTO}}(r, r', s, s') \triangleq \frac{1}{n-2} \sum_{i \in [n] \setminus \{r,s\}} k_\sigma(x_{i,1}, x_{r,r'}) k_\sigma(x_{i,2}, x_{s,s'}).$$

For ease of computation, we will rewrite $\hat{J}(w, \alpha)$ in terms of matrix operations. In what follows, we assume that $\widetilde{k}_\sigma(z_r, z_u) := \int k_\sigma(x, z_r) k_\sigma(x, z_u) dx$ has a closed-form expression or can otherwise be computed efficiently. Some examples [5] are give in Table 2.

Table 2: Some popular kernel functions and their associated $\widetilde{k}$. Here $\|\cdot\|_2$ is the Euclidean norm.

| Kernel | $k_\sigma(x, x')$ | $\widetilde{k}_\sigma(x, x')$ |
|---|---|---|
| Gaussian | $\left(\frac{1}{\sqrt{2\pi}\sigma}\right)^d \exp\left(-\frac{\|x-x'\|_2^2}{2\sigma^2}\right)$ | $k_{\sqrt{2}\sigma}(x, x')$ |
| Cauchy | $\left(\frac{1}{\sqrt{\pi}\sigma}\right)^d \left(\frac{\Gamma((1+d)/2)}{\Gamma(1/2)}\right) \left(\frac{\sigma^2 + \|x-x'\|_2^2}{\sigma^2}\right)^{-\frac{1+d}{2}}$ | $k_{2\sigma}(x, x')$ |
| Laplacian | $\frac{c_d}{\sigma^d} \exp\left(-\frac{\|x-x'\|_1}{\sigma}\right)$ | $\frac{1}{(4\sigma)^d} \prod_{l=1}^{d} \left(\frac{\sigma + |x_l - x'_l|}{\sigma}\right) \exp\left(-\frac{\|x-x'\|_1}{\sigma}\right)$ |

### S2.1 Full Optimization Problem

We begin by examining the first term of $\hat{J}(w, \alpha)$

$$\int q_{w,\alpha}(x, x')^2 dxdx' = \int \left( \sum_m w_m \sum_r \sum_{r'} \alpha_{m,r,r'} k_\sigma(x, x_{r,r'}) \sum_s \sum_{s'} \alpha_{m,s,s'} k_\sigma(x, x_{s,s'}) \right)$$

$$\times \left( \sum_j w_j \sum_u \sum_{u'} \alpha_{j,u,u'} k_\sigma(x, x_{u,u'}) \sum_v \sum_{v'} \alpha_{j,v,v'} k_\sigma(x, x_{v,v'}) \right) dxdx'$$

$$= \sum_{m,j} w_m w_j \sum_{r,r',u,u'} \sum_{s,s',v,v'} \alpha_{m,r,r'} \alpha_{m,s,s'} \alpha_{j,u,u'} \alpha_{j,v,v'}$$

$$\times \int k_\sigma(x, x_{r,r'}) k_\sigma(x, x_{u,u'}) dx \int k_\sigma(x', x_{s,s'}) k_\sigma(x', x_{v,v'}) dx'$$

$$= \sum_{m,j} w_m w_j \sum_{r,r',u,u'} \alpha_{m,r,r'} \alpha_{j,u,u'} \widetilde{k}_\sigma(x_{r,r'}, x_{u,u'}) \sum_{s,s',v,v'} \alpha_{m,s,s'} \alpha_{j,v,v'} \widetilde{k}_\sigma(x_{s,s'}, x_{v,v'})$$

$$= \sum_{m,j} w_m w_j \left( \alpha'_m G a_j \right)^2,$$

where $\times$ in the first line is scalar multiplication and $G$ is the kernel matrix of the data and is given by

$$G = \begin{bmatrix} \widetilde{k}_\sigma(x_{1,1}, x_{1,1}) & \widetilde{k}_\sigma(x_{1,1}, x_{1,2}) & \cdots & \cdots & \widetilde{k}_\sigma(x_{1,1}, x_{n,1}) & \widetilde{k}_\sigma(x_{1,1}, x_{n,2}) \\ \widetilde{k}_\sigma(x_{1,2}, x_{1,1}) & \widetilde{k}_\sigma(x_{1,2}, x_{1,2}) & \cdots & \cdots & \widetilde{k}_\sigma(x_{1,2}, x_{n,1}) & \widetilde{k}_\sigma(x_{1,2}, x_{n,2}) \\ \vdots & \vdots & \ddots & \ddots & \vdots & \vdots \\ \vdots & \vdots & \ddots & \ddots & \vdots & \vdots \\ \widetilde{k}_\sigma(x_{n,1}, x_{1,1}) & \widetilde{k}_\sigma(x_{n,1}, x_{1,2}) & \cdots & \cdots & \widetilde{k}_\sigma(x_{n,1}, x_{n,1}) & \widetilde{k}_\sigma(x_{n,1}, x_{n,2}) \\ \widetilde{k}_\sigma(x_{n,2}, x_{1,1}) & \widetilde{k}_\sigma(x_{n,2}, x_{1,2}) & \cdots & \cdots & \widetilde{k}_\sigma(x_{n,2}, x_{n,1}) & \widetilde{k}_\sigma(x_{n,2}, x_{n,2}) \end{bmatrix}.$$

Examining the second term of the ETISE yields

$$
\begin{aligned}
\mathbb{E}_q\left[q_{w,\alpha}\right] &\approx \sum_{m=1}^{M} \sum_{r=1}^{n} \sum_{r'=1}^{2} \sum_{s=1}^{n} \sum_{s'=1}^{2} w_m \alpha_{m,r,r'} \alpha_{m,s,s'} \hat{h}(r, r', s, s') \\
&= \begin{cases} \frac{1}{n-2} \sum_m \sum_{r,r'} \sum_{s,s'} w_m \alpha_{m,r,r'} \alpha_{m,s,s'} \sum_{i \in [n] \setminus \{r,s\}} k_\sigma(x_{i,1}, x_{r,r'}) k_\sigma(x_{i,2}, x_{s,s'}), & r \neq s \\ \frac{1}{n-1} \sum_m \sum_{r,r'} \sum_{s,s'} w_m \alpha_{m,r,r'} \alpha_{m,s,s'} \sum_{i \in [n] \setminus \{r\}} k_\sigma(x_{i,1}, x_{r,r'}) k_\sigma(x_{i,2}, x_{r,s'}), & r = s \end{cases} \\
&= \sum_{m=1}^{M} w_m \left(\alpha_m' C \alpha_m\right),
\end{aligned}
$$

where $C$ is given by

$$C = \begin{bmatrix} \hat{h}(1,1,1,1) & \hat{h}(1,1,1,2) & \cdots & \cdots & \hat{h}(1,1,n,1) & \hat{h}(1,1,n,2) \\ \hat{h}(1,2,1,1) & \hat{h}(1,2,1,2) & \cdots & \cdots & \hat{h}(1,2,n,1) & \hat{h}(1,2,n,2) \\ \vdots & \vdots & \ddots & \ddots & \vdots & \vdots \\ \vdots & \vdots & \ddots & \ddots & \vdots & \vdots \\ \hat{h}(n,1,1,1) & \hat{h}(n,1,1,2) & \cdots & \cdots & \hat{h}(n,1,n,1) & \hat{h}(n,1,n,2) \\ \hat{h}(n,2,1,1) & \hat{h}(n,2,1,2) & \cdots & \cdots & \hat{h}(n,2,n,1) & \hat{h}(n,2,n,2) \end{bmatrix}.$$

The diagonal blocks of size two of the matrix $C$ use the leave one out estimator, while the other entries use the leave two out estimator.

## S2.2 Coreset Approach

In the full problem the matrices $G, C \in \mathbb{R}^{2n \times 2n}$ grow linearly with the data. This can make the proposed optimization problem (6) costly to solve, as the complexity of gradient calculations are quadratic in the dimensions of $G, C$. Additionally, the complexity of evaluating out-of-sample data is quadratic in $n$ for general KDEs. The motivation of the coreset approach is to reduce this complexity.

KDEs traditionally center kernels at the location of each observation, i.e., $k_\sigma(\cdot, x_{i,i'})$, where we call $x_{i,i'}$ the kernel center. Rather than constraining the wKDE to have kernels centered at the observations, we can formulate the optimization problem with $R$ kernel centers $z_r \in \mathbb{R}^d$ for some suitably chosen $z_r$. Additionally, choosing $R \ll n$ will substantially reduce the complexity of gradient calculations and out-of-sample evaluation. The collection of kernel centers $z_r$ will be our coreset. We don't provide guarantees for the optimality of any particular coreset. The coreset could potentially be chosen as the cluster centers output by some clustering algorithm, some suitable subset of the data, or perhaps via some more principled scheme. In all of our experiments, we chose the coreset to be cluster centers output by mini-batch $k$-means, where the number of clusters was chosen to be $R > M$.

92 For the coreset approach, the ETISE has the same form but the matrices $G$ and $C$ have the form

$$
G = \begin{bmatrix}
\widetilde{k}_\sigma(z_1, z_1) & \cdots & \cdots & \widetilde{k}_\sigma(z_1, z_R) \\
\vdots & \ddots & \ddots & \vdots \\
\vdots & \ddots & \ddots & \vdots \\
\widetilde{k}_\sigma(z_R, z_1) & \cdots & \cdots & \widetilde{k}_\sigma(z_R, z_R)
\end{bmatrix},
$$

$$
C = \frac{1}{n}\sum_{i=1}^{n} C_i = \frac{1}{n}\sum_{i=1}^{n}
\begin{bmatrix}
k_\sigma(x_{i,1}, z_1)k_\sigma(x_{i,2}, z_1) & \cdots & \cdots & k_\sigma(x_{i,1}, z_1)k_\sigma(x_{i,2}, z_R) \\
\vdots & \ddots & \ddots & \vdots \\
\vdots & \ddots & \ddots & \vdots \\
k_\sigma(x_{i,1}, z_R)k_\sigma(x_{i,2}, z_1) & \cdots & \cdots & k_\sigma(x_{i,1}, z_R)k_\sigma(x_{i,2}, z_R)
\end{bmatrix}
$$

93 This is derived in the same way as the full problem, replacing the kernel centers $x_{r,r'}, x_{s,s'}$ with the
94 coreset $z_r$, and using the "leave-none-out" estimator in place of the LOO/LTO estimator $\hat{h}$.

## S2.3 Algorithm

96 Though the problem (7) is nonconvex, we observe that a properly initialized alternating pro-
97 jected stochastic gradient descent (APSGD) procedure produces good solutions in practice. Pseu-
98 docode for the APSGD algorithm for solving (7) is given in Algorithm 1. We mention that
99 the projections $\Pi_\Delta$ are onto the probability simplex, a decaying step size $\eta^{(t)}$ is used, and
100 stochasticity is introduced via the matrix $C^{(t)}$, which is a mini-batch version of $C$ defined by
101 $C_{a,b}^{(t)} = \frac{1}{|\Omega^{(t)}\setminus\{a,b\}|}\sum_{i\in|\Omega^{(t)}\setminus\{a,b\}|} k_\sigma(X_{i,1}, X_{\lfloor\frac{a}{2}\rfloor, a \bmod 2})k_\sigma(X_{i,2}, X_{\lfloor\frac{b}{2}\rfloor, b \bmod 2})$, where $\Omega^{(t)}$ is the
102 index set corresponding to the $t^{th}$ mini-batch.

---

**Algorithm 1** Alternating Projected SGD

---

1: **init:** $\alpha^{(0)}, w^{(0)}, \eta^{(0)}$
2: **procedure** APSGD$(\alpha^{(0)}, w^{(0)}, \eta^{(0)})$
3:     Form $G$
4:     **for** $t = 1, 2, \ldots$ **do**
5:         Take a minibatch of paired observations indexed by $\Omega^{(t)}$
6:         Form $C^{(t)}$ from the minibatch according to the definition of $C$
7:         $w^{(t)} = \Pi_\Delta(w^{(t-1)} - \eta^{(t)}\nabla_w\hat{J}(w^{(t-1)}, \alpha^{(t-1)}))$
8:         **for** $j = 1, \ldots, M$ **do**
9:             $\alpha_j^{(t)} = \Pi_\Delta(\alpha_j^{(t-1)} - \eta^{(t)}\nabla_{\alpha_j}\hat{J}(w^{(t)}, \alpha_j^{(t-1)}))$

---

## S2.4 Spectral Initialization

We adopt a spectral initialization scheme. First, the initialization is presented for the full problem, then we adapt it to the coreset approach. The idea here is, given some estimator of $q$, lets say $\tilde{q}$, to find a low rank approximation of $\tilde{q}$

$$
\tilde{q}(x, y) \approx \sum_{i=1}^{M} \lambda_i \psi_i(x)\psi_i(y)
$$

104 and then to use $\lambda_i$ and $\psi_i$ as starting points for our mixture weights and components. We do this by
105 using the full grouped sample data as an estimate of $q$ which we transform into a linear operator and
106 decompose using a functional eigenvector decomposition.

107 We begin with a standard KDE applied to our full samples using a product kernel:

$$
f_\sigma(y, y') = \frac{1}{2n}\sum_{i=1}^{n} k_\sigma(y, x_{i,1})k_\sigma(y', x_{i,2}) + k_\sigma(y, x_{i,2})k_\sigma(y', x_{i,1}).
$$

108 Note that we include centers at both $(x_{i,1}, x_{i,2})$ and $(x_{i,2}, x_{i,1})$ so our KDE is symmetric in $y, y'$.

By Lemmas 5.1 and 8.2 of Vandermeulen and Scott [12], $f_\sigma$ can be viewed as an element of a tensor product space $L^2(\mathbb{R}^d) \otimes L^2(\mathbb{R}^d)$ as follows

$$f_\sigma = \frac{1}{2n} \sum_{i=1}^n k_\sigma(\cdot, x_{i,1}) \otimes k_\sigma(\cdot, x_{i,2}) + k_\sigma(\cdot, x_{i,2}) \otimes k_\sigma(\cdot, x_{i,1}).$$

By the Lemmas referenced above, there is a unitary transformation on the KDE $f_\sigma$ such that it can be viewed as a linear operator $T : L^2(\mathbb{R}^d) \to L^2(\mathbb{R}^d)$ given by

$$T(g) := \sum_{i=1}^n k_\sigma(\cdot, x_{i,1})\langle k_\sigma(\cdot, x_{i,2}), g(\cdot)\rangle_{L^2} + k_\sigma(\cdot, x_{i,2})\langle k_\sigma(\cdot, x_{i,1}), g(\cdot)\rangle_{L^2}, \quad \forall g \in L^2(\mathbb{R}^d)$$

which is a symmetric operator since it includes both $k_\sigma(\cdot, x_{i,1})\langle k_\sigma(\cdot, x_{i,2}), g(\cdot)\rangle_{L^2}$ and $k_\sigma(\cdot, x_{i,2})\langle k_\sigma(\cdot, x_{i,1}), g(\cdot)\rangle_{L^2}$ terms. We have removed the $1/(2n)$ coefficient since it will not affect the spectral decomposition. For any $g \in L^2$ the quantity $\langle k_\sigma(\cdot, x_{i,i'}), g(\cdot)\rangle_{L^2}$ will be a finite scalar, so $T(g)$ will be a linear combination of the $k_\sigma(\cdot, x_{i,i'})$. Therefore, eigenvectors of the above linear operator will have the form $g(\cdot) = \sum_{j,j'} \beta_{j,j'} k_\sigma(\cdot, x_{j,j'})$ since $T$ applied to any vector must lie in the span of $k_\sigma(\cdot, x_{j,j'})$. Evaluating $T$ on vectors of this form (not necessarily an eigenvector) will yield

$$T(g) := \sum_{i=1}^n \Big\{ k_\sigma(\cdot, x_{i,1})\langle k_\sigma(\cdot, x_{i,2}), \sum_{j,j'} \beta_{j,j'} k_\sigma(\cdot, x_{j,j'})\rangle_{L^2}$$

$$ + k_\sigma(\cdot, x_{i,2})\langle k_\sigma(\cdot, x_{i,1}), \sum_{j,j'} \beta_{j,j'} k_\sigma(\cdot, x_{j,j'})\rangle_{L^2} \Big\}$$

$$ = \sum_i \zeta_{i,1} k_\sigma(\cdot, x_{i,1}) + \zeta_{i,2} k_\sigma(\cdot, x_{i,2})$$

where $\zeta_{i,1} = \sum_{j,j'} \beta_{j,j'} \tilde{k}_\sigma(x_{i,2}, x_{j,j'})$, and $\zeta_{i,2} = \sum_{j,j'} \beta_{j,j'} \tilde{k}_\sigma(x_{i,1}, x_{j,j'})$ and $\tilde{k}_\sigma(y, y') := \langle k_\sigma(\cdot, y), k_\sigma(\cdot, y')\rangle_{L^2}$.

Define the ordering of the elements of $\beta$ and $\zeta$ by

$$\beta = [\beta_{1,1}, \beta_{1,2}, \beta_{2,1}, \beta_{2,2}, \dots, \beta_{n,1}, \beta_{n,2}]',$$
$$\zeta = [\zeta_{1,1}, \zeta_{1,2}, \zeta_{2,1}, \zeta_{2,2}, \dots, \zeta_{n,1}, \zeta_{n,2}]'.$$

Then we have

$$\zeta = \bar{G}\beta,$$

where

$$\bar{G} = \begin{bmatrix}
\tilde{k}_\sigma(x_{1,2}, x_{1,1}) & \tilde{k}_\sigma(x_{1,2}, x_{1,2}) & \tilde{k}_\sigma(x_{1,2}, x_{2,1}) & \cdots & \tilde{k}_\sigma(x_{1,2}, x_{n,1}) & \tilde{k}_\sigma(x_{1,2}, x_{n,2}) \\
\tilde{k}_\sigma(x_{1,1}, x_{1,1}) & \tilde{k}_\sigma(x_{1,1}, x_{1,2}) & \tilde{k}_\sigma(x_{1,1}, x_{2,1}) & \cdots & \tilde{k}_\sigma(x_{1,1}, x_{n,1}) & \tilde{k}_\sigma(x_{1,1}, x_{n,2}) \\
\tilde{k}_\sigma(x_{2,2}, x_{1,1}) & \tilde{k}_\sigma(x_{2,2}, x_{1,2}) & \tilde{k}_\sigma(x_{2,2}, x_{2,1}) & \cdots & \tilde{k}_\sigma(x_{2,2}, x_{n,1}) & \tilde{k}_\sigma(x_{2,2}, x_{n,2}) \\
\tilde{k}_\sigma(x_{2,1}, x_{1,1}) & \tilde{k}_\sigma(x_{2,1}, x_{1,2}) & \tilde{k}_\sigma(x_{2,1}, x_{2,1}) & \cdots & \tilde{k}_\sigma(x_{2,1}, x_{n,1}) & \tilde{k}_\sigma(x_{2,1}, x_{n,2}) \\
\vdots & \vdots & \vdots & \ddots & \vdots & \vdots \\
\tilde{k}_\sigma(x_{n,2}, x_{1,1}) & \tilde{k}_\sigma(x_{n,2}, x_{1,2}) & \tilde{k}_\sigma(x_{n,2}, x_{2,1}) & \cdots & \tilde{k}_\sigma(x_{n,2}, x_{n,1}) & \tilde{k}_\sigma(x_{n,2}, x_{n,2}) \\
\tilde{k}_\sigma(x_{n,1}, x_{1,1}) & \tilde{k}_\sigma(x_{n,1}, x_{1,2}) & \tilde{k}_\sigma(x_{n,1}, x_{2,1}) & \cdots & \tilde{k}_\sigma(x_{n,1}, x_{n,1}) & \tilde{k}_\sigma(x_{n,1}, x_{n,2})
\end{bmatrix}$$

is a row permutation of $G$ defined for the full problem, obtained by exchanging rows corresponding to the first and second elements of each paired observation. The takeaway is that the coefficients of the eigenvectors of $T$ are given by the right eigenvectors of $\bar{G}$. In particular, we will take the right eigenvectors of $\bar{G}$ corresponding to the $M$ largest real eigenvalues with the intuition that they will capture the dominant modes of $T$. Note that these eigenvectors will contain real-valued entries by the spectral theorem since $T$ is symmetric. We call these eigenvectors the first, second, and so on. It should be noted that $\bar{G}$ is not a symmetric matrix and so eigenvectors should be found according to, for example, the power iteration or orthogonal iteration. In general the eigenvectors of $\bar{G}$ will have negative entries and not sum to one, so we project these eigenvectors onto the probability simplex to obtain the non-negative weights for initialization where we take $\alpha_1$ to the be the projection of the first eigenvector of $\bar{G}$, $\alpha_2$ to the be the second, and so on. For the initial $w_i$, We take $w_1$ to be the first eigenvalue of $\bar{G}$, $w_2$ to be the second and so on, projecting the resulting $w = [w_1, w_2, \dots, w_M]'$ onto the probability simplex.

136 **Coreset Approach Initialization** Initialization for the coreset approach is similar. Since we assume
137 no relationship between $z_i$, we don't have the same notion of using paired kernel centers even though
138 our data is still paired. However, we can still write the KDE using a product kernel over the coreset as

$$f_\sigma(y, y') = \frac{1}{R} \sum_{i=1}^{R} k_\sigma(y, z_i) k_\sigma(y', z_i).$$

This KDE is symmetric since the same kernel center is used in each term of the product kernel. Again
appealing to Lemmas 5.1 and 8.2 of Vandermeulen and Scott [12], $f_\sigma$ can be viewed as an element of
a tensor product space $L^2(\mathbb{R}^d) \otimes L^2(\mathbb{R}^d)$ as follows

$$f_\sigma = \frac{1}{R} \sum_{i=1}^{R} k_\sigma(\cdot, z_i) \otimes k_\sigma(\cdot, z_i).$$

139 By the Lemmas referenced above, there is a unitary transformation on the KDE $f_\sigma$ such that it can be
140 viewed as a linear operator $T : L^2(\mathbb{R}^d) \to L^2(\mathbb{R}^d)$ given by

$$T(g) = \sum_{i=1}^{R} k_\sigma(\cdot, z_i) \langle k_\sigma(\cdot, z_i), g(\cdot) \rangle_{L^2}, \quad \forall g \in L^2.$$

141 By the same argument as the full problem, the eigenvectors of the above operator have the form
142 $g(\cdot) = \sum_j \beta_j k_\sigma(\cdot, z_j)$. Applying $T$ to a vector of this form (not necessarily an eigenvector), we have

$$T(\sum_j \beta_j k_\sigma(\cdot, z_j)) = \sum_{i=1}^{R} k_\sigma(\cdot, z_i) \langle k_\sigma(\cdot, z_i), \sum_j \beta_j k_\sigma(\cdot, z_j) \rangle_{L^2}$$

$$= \sum_{i=1}^{R} \zeta_i k_\sigma(\cdot, z_i),$$

143 where $\zeta_i = \sum_j \beta_j k_\sigma(\cdot, z_i)$. In this setting we have the standard ordering, $\beta = [\beta_1, \beta_2, \ldots, \beta_R]'$ and
144 $\zeta = [\zeta_1, \zeta_2, \ldots, \zeta_R]'$. In matrix form, the relationship between $\beta$ and $\zeta$ is given by

$$\zeta = G\beta,$$

145 where $G$ is as previously defined for the coreset approach. No row permutation is needed in this
146 setting as both centers of our product kernel are the same. From this point, the initialization scheme
147 is essentially the same as for the full problem, but using the eigenvectors and eigenvalues of $G$. One
148 key difference is that $G$ *is* a symmetric matrix, so the eigenvalues and eigenvectors of $G$ can be found
149 using any standard solver.

## S3  Proof and General Form of Theorem 1

151 Theorem 1 was presented in the main paper for groups of size $N = 2$. Here, we provide the proof for
152 groups of size two, as well as the proof for the more general case of arbitrary group size $N > 2$. One
153 tool we will use is Hoeffding's inequality for independent bounded random variables, which we state
154 here for completeness.

155 **Theorem.** *Hoeffding's Inequality: Let $V_1, V_2, \ldots, V_n$ be independent bounded random variables*
156 *such that $a_i \leq V_i \leq b_i$ with probability one. If $S_n = \sum_{i=1}^{n} V_i$, then for all $t > 0$*

$$P\left\{ \left| S_n - \mathbb{E}\{S_n\} \right| \geq t \right\} \leq 2 \exp\left\{ -\frac{2t^2}{\sum_{i=1}^{n}(b_i - a_i)^2} \right\}.$$

### S3.1  Proof of Theorem 1: Groups of Size Two

158 We restate Theorem 1 for convenience.

159 **Theorem 1.** *Let $\epsilon > 0$ and set $\delta = 8(n^2 - n) \exp\{-\frac{\sigma^{4d}(n-2)\epsilon^2}{8C_k^4}\} + 8n \exp\{-\frac{\sigma^{4d}(n-1)\epsilon^2}{8C_k^4}\}$. With*
160 *probability at least $1 - \delta$ the following holds:*

$$\|q - q_{\hat{w}, \hat{\alpha}}\|_2^2 \leq \inf_{w \in \Delta^M, \ \alpha \in \Delta_M^{2n}} \|q - q_{w, \alpha}\|_2^2 + \epsilon.$$

*Proof.* Our goal is to bound $|J(w, a) - \hat{J}(w, a)|$ uniformly over $w \in \Delta^M$, $\alpha \in \Delta_M^{2n}$. Recall the following definitions

$$h(r, r', s, s') := \int k_\sigma(x, x_{r,r'}) k_\sigma(x', x_{s,s'}) q(x, x') dx dx'$$

$$\hat{h}(r, r', s, s') := \begin{cases} \hat{h}_{\text{LTO}}(r, r', s, s'), & r \neq s \\ \hat{h}_{\text{LOO}}(r, r', s'), & r = s \end{cases}$$

$$\hat{h}_{\text{LOO}}(r, r', r'') := \frac{1}{n-1} \sum_{i \in [n] \setminus \{r\}} k_\sigma(x_{i,1}, x_{r,r'}) k_\sigma(x_{i,2}, x_{r,r''})$$

$$\hat{h}_{\text{LTO}}(r, r', s, s') := \frac{1}{n-2} \sum_{i \in [n] \setminus \{r,s\}} k_\sigma(x_{i,1}, x_{r,r'}) k_\sigma(x_{i,2}, x_{s,s'}).$$

The use of the leave one out (LOO) and leave two out (LTO) estimators above is to ensure independence so that we will be able to apply Hoeffding's inequality. We have

$$P_q \Big\{ \sup_{\substack{w \in \Delta^M \\ \alpha \in \Delta_M^{2n}}} |J(w, \alpha) - \hat{J}(w, \alpha)| > \frac{\epsilon}{2} \Big\}$$

$$= P_q \Big\{ \sup_{\substack{w \in \Delta^M \\ \alpha \in \Delta_M^{2n}}} \Big| \sum_{m=1}^M w_m \sum_{r=1}^n \sum_{s=1}^n \sum_{r'=1}^2 \sum_{s'=1}^2 \alpha_{m,r,r'} \alpha_{m,s,s'} h(r, r', s, s')$$

$$- \sum_{m=1}^M w_m \sum_{r=1}^n \sum_{s=1}^n \sum_{r'=1}^2 \sum_{s'=1}^2 \alpha_{m,r,r'} \alpha_{m,s,s'} \hat{h}(r, r', s, s') \Big| > \frac{\epsilon}{4} \Big\}$$

$$\leq P_q \Big\{ \sup_{\substack{w \in \Delta^M \\ \alpha \in \Delta_M^{2n}}} \sum_m \sum_{r,s} \sum_{r',s'} w_m \alpha_{m,r,r'} \alpha_{m,s,s'} \Big| h(r, r', s, s') - \hat{h}(r, r', s, s') \Big| > \frac{\epsilon}{4} \Big\}$$

$$\leq P_q \Big\{ \max_{r,s,r',s'} \Big| h(r, r', s, s') - \hat{h}(r, r', s, s') \Big| > \frac{\epsilon}{4} \Big\}$$

$$\leq \sum_{r,s} \sum_{r',s'} P_q \Big\{ \Big| h(r, r', s, s') - \hat{h}(r, r', s, s') \Big| > \frac{\epsilon}{4} \Big\}.$$

The second step above is due to the triangle inequality, and the penultimate step is due to simplex constraints on $w, \alpha$. Let $k_i(r, r', s, s') := k_\sigma(x_{i,1}, x_{r,r'}) k_\sigma(x_{i,2}, x_{s,s'})$. Noting that $h(r, r', s, s') = \mathbb{E}_{(x_{i,1}, x_{i,2}) \sim q} \{ k_i(r, r', s, s') \}$,

$$P_q \Big\{ |h(r, r', s, s') - \hat{h}(r, r', s, s')| > \frac{\epsilon}{4} \Big\}$$

$$= \begin{cases} P_q \Big\{ \Big| \mathbb{E}_{(x_{i,1}, x_{i,2}) \sim q} \{ k_i(r, r', s, s') \} - \frac{1}{n-2} \sum_{i \in [n] \setminus \{r,s\}} k_i(r, r', s, s') \Big| > \frac{\epsilon}{4} \Big\}, & r \neq s \\ P_q \Big\{ \Big| \mathbb{E}_{(x_{i,1}, x_{i,2}) \sim q} \{ k_i(r, r', s, s') \} - \frac{1}{n-1} \sum_{i \in [n] \setminus \{r\}} k_i(r, r', s, s') \Big| > \frac{\epsilon}{4} \Big\}, & r = s \end{cases}$$

$$= \begin{cases} P_q \Big\{ \Big| \frac{1}{n-2} \sum_{i \in [n] \setminus \{r,s\}} \mathbb{E}_{(x_{i,1}, x_{i,2}) \sim q} \{ k_i(r, r', s, s') \} - k_i(r, r', s, s') \Big| > \frac{\epsilon}{4} \Big\}, & r \neq s \\ P_q \Big\{ \Big| \frac{1}{n-1} \sum_{i \in [n] \setminus \{r\}} \mathbb{E}_{(x_{i,1}, x_{i,2}) \sim q} \{ k_i(r, r', s, s') \} - k_i(r, r', s, s') \Big| > \frac{\epsilon}{4} \Big\}, & r = s \end{cases}$$

$$= \begin{cases} P_q \Big\{ \Big| \sum_{i \in [n] \setminus \{r,s\}} \mathbb{E}_{(x_{i,1}, x_{i,2}) \sim q} \{ k_i(r, r', s, s') \} - k_i(r, r', s, s') \Big| > \frac{(n-2)\epsilon}{4} \Big\}, & r \neq s \\ P_q \Big\{ \Big| \sum_{i \in [n] \setminus \{r\}} \mathbb{E}_{(x_{i,1}, x_{i,2}) \sim q} \{ k_i(r, r', s, s') \} - k_i(r, r', s, s') \Big| > \frac{(n-1)\epsilon}{4} \Big\}, & r = s \end{cases}$$

The terms $k_i(r, r', s, s')$ are independent random variables due to use of the LOO/LTO estimator. By assumption, $0 \leq k_i(r, r', s, s') \leq C_k^2 \sigma^{-2d}$ so the $k_i$ are bounded for fixed $\sigma > 0$. We apply

Hoeffding's inequality

$$P_q\left\{\left|h(r,r',s,s') - \hat{h}(r,r',s,s')\right| > \frac{\epsilon}{4}\right\} \leq \begin{cases} 2\exp\{-\frac{2(n-2)^2\epsilon^2}{16(n-2)C_k^4\sigma^{-4d}}\}, & r \neq s \\ 2\exp\{-\frac{2(n-1)^2\epsilon^2}{16(n-1)C_k^4\sigma^{-4d}}\}, & r = s \end{cases}$$

$$\leq \begin{cases} 2\exp\{-\frac{\sigma^{4d}(n-2)\epsilon^2}{8C_k^4}\}, & r \neq s \\ 2\exp\{-\frac{\sigma^{4d}(n-1)\epsilon^2}{8C_k^4}\}, & r = s \end{cases}. \quad \text{(S1)}$$

Substituting backward we obtain the desired upper bound

$$P_q\{\sup_{\substack{w\in\Delta^M \\ \alpha\in\Delta_M^{2n}}} |J(w,\alpha) - \hat{J}(w,\alpha)| > \frac{\epsilon}{2}\} \leq \sum_{r,s}\sum_{r',s'} \begin{cases} 2\exp\{-\frac{\sigma^{4d}(n-2)\epsilon^2}{8C_k^4}\}, & r \neq s \\ 2\exp\{-\frac{\sigma^{4d}(n-1)\epsilon^2}{8C_k^4}\}, & r = s \end{cases}$$

$$= 8(n^2 - n)\exp\{-\frac{\sigma^{4d}(n-2)\epsilon^2}{8C_k^4}\} + 8n\exp\{-\frac{\sigma^{4d}(n-1)\epsilon^2}{8C_k^4}\}.$$

Letting $\delta = 8(n^2 - n)\exp\{-\frac{\sigma^{4d}(n-2)\epsilon^2}{8C_k^4}\} + 8n\exp\{-\frac{\sigma^{4d}(n-1)\epsilon^2}{8C_k^4}\}$, we have

$$J(w,\alpha) - \frac{\epsilon}{2} \leq \hat{J}(w,\alpha) \leq J(w,\alpha) + \frac{\epsilon}{2} \quad \forall w, \alpha \quad \text{(S2)}$$

with probability at least $1 - \delta$. Thus, with probability at least $1 - \delta$, for any $w \in \Delta^M, \alpha \in \Delta_M^{2n}$

$$J(\hat{w}, \hat{\alpha}) \leq \hat{J}(\hat{w}, \hat{\alpha}) + \frac{\epsilon}{2}$$

$$\leq \hat{J}(w, \alpha) + \frac{\epsilon}{2}$$

$$\leq J(w, \alpha) + \epsilon,$$

where $\hat{w}, \hat{\alpha}$ are defined in (6). Then with probability at least $1 - \delta$

$$\hat{J}(\hat{w}, \hat{\alpha}) \leq \inf_{\substack{w\in\Delta^M \\ \alpha\in\Delta_M^{2n}}} J(w,\alpha) + \epsilon. \quad \text{(S3)}$$

Combining (S3) with the definition of the ISE shows, with probability at least $1 - \delta$,

$$\|q - q_{\hat{w},\hat{\alpha}}\|_2^2 \leq \inf_{\substack{w\in\Delta^M \\ \alpha\in\Delta_M^{2n}}} \|q - q_{w,\alpha}\|_2^2 + \epsilon.$$

$\square$

## S3.2 Theorem 1: Arbitrary Group Size

### S3.2.1 Preliminaries

Before beginning the proof, we start by redefining $q$, $q_{w,a}$, $J$, and $\hat{J}$ for arbitrary group size. Once this is done, the proof will follow the same basic steps as the proof for groups of size two.

Suppose we change the problem setup only in the size of the grouped observations. Consider grouped observations of size $N$. Consider a set of $n$ grouped observations $\mathbf{x}_1, \ldots, \mathbf{x}_n$ with $\mathbf{x}_i = (x_{i,1}, \ldots, x_{i,N}) \in \mathbb{R}_N^d := \underbrace{\mathbb{R}^d \times \ldots \times \mathbb{R}^d}_{N}$ drawn i.i.d. from

$$q(y_1, y_2, \ldots, y_N) = \sum_{m=1}^M w_m^* p_m^*(y_1) p_m^*(y_2) \ldots p_m^*(y_N), \quad y_1, y_2, \ldots, y_N \in \mathbb{R}^d. \quad \text{(S4)}$$

Similar to the paired observation setting, a wKDE in this setting will have the form

$$p(y; \theta) = \sum_{r=1}^n \sum_{r'=1}^N \theta_{r,r'} k_\sigma(y, x_{r,r'}).$$

We may write the corresponding estimator of $q$

$$q_{w,\alpha}(y_1, y_2, \ldots, y_N) = \sum_{m=1}^{M} w_m p(y_1; \alpha_m) p(y_2; \alpha_m) \ldots p(y_N; \alpha_m)$$

where $\alpha_m = [\alpha_{m,1,1} \ldots \alpha_{m,1,N} \ldots \ldots \alpha_{m,n,1} \ldots \alpha_{m,n,N}]' \in \Delta^{Nn}$ for $m = 1, \ldots, M$, with $\alpha_{m,r,r'}$ corresponding to the weight of the kernel centered at $x_{r,r'}$ in the estimate of the $m^{th}$ mixture component.

In what follows we use $\sum_{r,r'} := \sum_{r_1,r_1'} \ldots \sum_{r_N,r_N'}$ to ease notation. Similar to the paired sample case, we define

$$J(w, \alpha) := \int q_{w,\alpha}^2(y_1, \ldots, y_N) dy_1 \ldots dy_N - 2 \sum_{m,r,r'} \left( \prod_{i \in [N]} w_m \alpha_{m,r_i,r_i'} \right) h(r_1, r_1', \ldots, r_N, r_N')$$

$$\hat{J}(w, \alpha) := \int q_{w,\alpha}^2(y_1, \ldots, y_N) dy_1 \ldots dy_N - 2 \sum_{m,r,r'} \left( \prod_{i \in [N]} w_m \alpha_{m,r_i,r_i'} \right) \hat{h}(r_1, r_1', \ldots, r_N, r_N'),$$

where

$$h(r_1, r_1', \ldots, r_N, r_N') := \int k_\sigma(y_1, x_{r_1,r_1'}) \ldots k_\sigma(y_N, x_{r_N,r_N'}) q(y_1, \ldots, y_N) dy_1 \ldots dy_N,$$

$$\hat{h} := \hat{h}_{\text{LNO}}(r_1, r_1', \ldots, r_N, r_N') := \frac{1}{n-N} \sum_{i \in [n] \setminus L(r_1, r_2, \ldots, r_N)} k_\sigma(x_{i,1}, x_{r_1,r_1'}) \ldots k_\sigma(x_{i,N}, x_{r_N,r_N'}),$$

where $L(r_1, r_2, \ldots, r_N)$ is any subset of $[n]$ containing $\{r_1, r_2, \ldots, r_N\}$ and having cardinality $N$. If $r_1, r_2, \ldots, r_N$ are not distinct, the additional indices can be chosen arbitrarily. For simplicity, we use a leave-N-out (LNO) estimator $\hat{h}_{\text{LNO}}$ rather than a hybrid estimator like we used in the case of paired observations. As in the paired observation setting, we define

$$(\hat{w}, \hat{\alpha}) := \underset{w \in \Delta^M, \, \alpha \in \Delta_M^{Nn}}{\arg\min} \hat{J}(w, \alpha),$$

and similarly define $\hat{q} := q_{\hat{w},\hat{\alpha}}$. Whenever $\hat{w}, \hat{\alpha}, q$, or $q_{\hat{w},\hat{\alpha}}$ are referenced in the arbitrary group size setting, we will be referring to these estimators.

## S3.2.2   Proof of Theorem 1: Arbitrary Group Size

We now state Theorem 1 for arbitrary group size.

**Theorem 1a.** *Given grouped observations of size $N$, let $\epsilon > 0$ and set $\delta = 2(Nn)^N \exp\left\{-\frac{\sigma^{2Nd}(n-N)\epsilon^2}{8C_k^{2N}}\right\}$. With probability at least $1 - \delta$ the following holds:*

$$\|q - q_{\hat{w},\hat{\alpha}}\|_2^2 \leq \inf_{w \in \Delta^M, \, \alpha \in \Delta_M^{Nn}} \|q - q_{w,\alpha}\|_2^2 + \epsilon.$$

*Proof.* The proof proceeds as in the paired observation setting. In particular,

$$P_q\left\{ \sup_{\substack{w \in \Delta^M \\ \alpha \in \Delta_M^{Nn}}} |J(w, \alpha) - \hat{J}(w, \alpha)| > \frac{\epsilon}{2} \right\}$$

$$\leq P_q\left\{ \sup_{\substack{w \in \Delta^M \\ \alpha \in \Delta_M^{Nn}}} \sum_{m,r,r'} \prod_{i \in [N]} w_m \alpha_{m,r_i,r_i'} \left| h(r_1, r_1', \ldots, r_N, r_N') - \hat{h}(r_1, r_1', \ldots, r_N, r_N') \right| > \frac{\epsilon}{4} \right\}$$

$$\leq P_q\left\{ \max_{r_1,r_1',\ldots,r_N,r_N'} \left| h(r_1, r_1', \ldots, r_N, r_N') - \hat{h}(r_1, r_1', \ldots, r_N, r_N') \right| > \frac{\epsilon}{4} \right\}$$

$$\leq \sum_{r,r'} P_q\left\{ \left| h(r_1, r_1', \ldots, r_N, r_N') - \hat{h}(r_1, r_1', \ldots, r_N, r_N') \right| > \frac{\epsilon}{4} \right\}$$

203 The first step above is due to the triangle inequality, and the penultimate step is due to simplex con-
204 straints on $w, \alpha$. Let $k_i(r_1, r_1', \ldots, r_N, r_N') := k_\sigma(x_{i,1}, x_{r_1, r_1'}) k_\sigma(x_{i,2}, x_{r_2, r_2'}) \cdots k_\sigma(x_{i,N}, x_{r_N, r_N'})$.
205 Noting that $h(r_1, r_1', \ldots, r_N, r_N') = \mathbb{E}_q\{k_\sigma(x_{i,1}, x_{r_1, r_1'}) \cdots k_\sigma(x_{i,N}, x_{r_N, r_N'})\}$, we have

$$P_q\left\{\left|h(r_1, r_1', \ldots, r_N, r_N') - \hat{h}(r_1, r_1', \ldots, r_N, r_N')\right| > \frac{\epsilon}{4}\right\}$$

$$= P_q\left\{\left|\sum_{i \in [n] \backslash L(r_1, \ldots, r_N)} \mathbb{E}_{(x_{i,1}, \ldots, x_{i,N}) \sim q}\{k_i(r_1, r_1', \ldots, r_N, r_N')\}\right.\right.$$

$$\left.\left. - k_i(r_1, r_1', \ldots, r_N, r_N')\right| > \frac{(n-N)\epsilon}{4}\right\}$$

206 The terms $k_i(r_1, r_1', \ldots, r_N, r_N')$ are independent random variables due to use of the LNO estimator.
207 By assumption, $0 \le k_i(r_1, r_1', \ldots, r_N, r_N') \le C_k^N \sigma^{-Nd}$ so the $k_i$ are bounded for fixed $\sigma > 0$. We
208 apply Hoeffding's inequality

$$P_q\left\{\left|h(r_1, r_1', \ldots, r_N, r_N') - \hat{h}(r_1, r_1', \ldots, r_N, r_N')\right| > \frac{\epsilon}{4}\right\} \le 2\exp\{-\frac{2(n-N)^2\epsilon^2}{16(n-N)C_k^{2N}\sigma^{-2Nd}}\}$$

$$= 2\exp\{-\frac{\sigma^{2Nd}(n-N)\epsilon^2}{8C_k^{2N}}\}.$$

209 Substituting backward we obtain the desired upper bound

$$P_q\{\sup_{\substack{w \in \Delta^M \\ \alpha \in \Delta_M^{Nn}}} |J(w, \alpha) - \hat{J}(w, \alpha)| > \frac{\epsilon}{2}\} \le \sum_{r_1, r_1'} \cdots \sum_{r_N, r_N'} 2\exp\{-\frac{\sigma^{2Nd}(n-N)\epsilon^2}{8C_k^{2N}}\}$$

$$= 2(Nn)^N \exp\{-\frac{\sigma^{2Nd}(n-N)\epsilon^2}{8C_k^{2N}}\}$$

210 From here the proof is identical to the paired observation case, but with

$$\delta = 2(Nn)^N \exp\{-\frac{\sigma^{2Nd}(n-N)\epsilon^2}{8C_k^{2N}}\}.$$

211 □

## S4  Theorem 2

213 In this section we give the proof of Theorem 2 for groups of size two, before extending it to groups
214 of arbitrary size. For readability, we first present some intermediate results to be used in the main
215 proofs.

### S4.1  Intermediate Results

217 We first prove two supporting results.

218 **Lemma 1.** *For any $1 \le p < \infty$, any $f, g \in L^p$, and any integer $a \ge 2$,*

$$\|f^{\times a} - g^{\times a}\|_p \le \|f\|_p^{a-1} \|f - g\|_p + \|g\|_p \|f^{\times(a-1)} - g^{\times(a-1)}\|_p,$$

219 *where $f^{\times a}(y_1, y_2, \ldots, y_a) := f(y_1) f(y_2) \cdots f(y_a)$.*

*Proof.* Let $f, g \in L^p, 1 \le p < \infty$. Then

$$
\begin{aligned}
\|f^{\times a} - g^{\times a}\|_p &= \|f^{\times a} - f^{\times(a-1)} \times g + f^{\times(a-1)} \times g - g^{\times a}\|_p \\
&\le \|f^{\times a} - f^{\times(a-1)} \times g\|_p + \|f^{\times(a-1)} \times g - g^{\times a}\|_p \\
&= \left( \int |f(x_1) \cdots f(x_{a-1})(f(x_a) - g(x_a))|^p dx_1 \ldots dx_a \right)^{\frac{1}{p}} \\
&\quad + \left( \int |g(x_a)(f(x_1) \cdots f(x_{a-1}) - g(x_1) \cdots g(x_{a-1}))|^p dx_1 \ldots dx_a \right)^{\frac{1}{p}} \\
&= \|f^{\times(a-1)}\|_p \|f - g\|_p + \|g\|_p \|f^{\times(a-1)} - g^{\times(a-1)}\|_p \\
&= \|f\|_p^{a-1} \|f - g\|_p + \|g\|_p \|f^{\times(a-1)} - g^{\times(a-1)}\|_p.
\end{aligned}
$$

$\square$

We have the following corollary.

**Corollary 1.** *For any $1 \le p < \infty$, any $f, g \in L^p$, and any integer $a \ge 2$,*

$$
\|f^{\times a} - g^{\times a}\|_p \le \left( \sum_{b=1}^{a} \|f\|_p^{a-b} \|g\|_p^{b-1} \right) \|f - g\|_p.
$$

*Proof.* The proof is by induction. Lemma 1 provides the base of the recursion for $a = 2$. Now suppose the statement is true for $a \ge 2$. To prove the statement for $a + 1$, we apply Lemma 1 again, together with the induction hypothesis, to get

$$
\begin{aligned}
\|f^{\times(a+1)} - g^{\times(a+1)}\|_p &\le \|f\|_p^a \|f - g\|_p + \|g\|_p \|f^{\times a} - g^{\times a}\|_p \\
&\le \|f\|_p^a \|f - g\|_p + \|g\|_p \left( \sum_{b=1}^{a} \|f\|_p^{a-b} \|g\|_p^{b-1} \right) \|f - g\|_p \\
&= \left( \sum_{b=1}^{a+1} \|f\|_p^{a+1-b} \|g\|_p^{b-1} \right) \|f - g\|_p.
\end{aligned}
$$

This completes the proof. $\square$

## S4.2 Proof of Theorem 2: Groups of Size Two

We restate Theorem 2 for convenience.

**Theorem 2.** *If $\sigma \to 0$ and $\frac{n\sigma^{4d}}{\log n} \to \infty$ as $n \to \infty$, then $\|q - q_{\hat{w}, \hat{\alpha}}\|_1 \xrightarrow{a.s.} 0$.*

*Proof.* Lemma 3.1 of [3] states that if $\int \hat{q} = 1$ and $\|\hat{q} - q\|_2 \xrightarrow{a.s.} 0$, then $\|\hat{q} - q\|_1 \xrightarrow{a.s.} 0$. Since $\int \hat{q} = 1$ in our case, our strategy is to show $\|\hat{q} - q\|_2 \xrightarrow{a.s.} 0$. To do this it suffices to show that

$$
\|q - \hat{q}\|_2^2 - \inf_{\substack{w \in \Delta^M \\ \alpha \in \Delta_M^{Nn}}} \|q - q_{w,\alpha}\|_2^2 \xrightarrow{a.s.} 0 \tag{S5}
$$

and

$$
\inf_{\substack{w \in \Delta^M \\ \alpha \in \Delta_M^{Nn}}} \|q - q_{w,\alpha}\|_2 \xrightarrow{a.s.} 0. \tag{S6}
$$

To show (S5), by the Borel-Cantelli lemma, it suffices to show that for all $\epsilon > 0$,

$$
\sum_{n=1}^{\infty} P_q \left( \|q - \hat{q}\|_2^2 - \inf_{\substack{w \in \Delta^M \\ \alpha \in \Delta_M^{2n}}} \|q - q_{w,\alpha}\|_2^2 \ge \epsilon \right) < \infty.
$$

Thus let $\epsilon > 0$. By Theorem 1a, the probability in question is at most

$$\delta = 2(nN)^N \exp\left\{-\frac{(n-N)\sigma^{2Nd}\epsilon^2}{8C_k^{2N}}\right\}$$

$$= 8\exp\left\{-2\log n\left(\frac{(n-2)\sigma^{4d}\epsilon^2}{16C_k^4\log n} - 1\right)\right\}.$$

By assumption on the growth of $n$ and $\sigma$, there exists $N_\epsilon$ such that for all $n \geq N_\epsilon$,

$$\frac{(n-2)\sigma^{4d}\epsilon^2}{16C_k^4\log n} \geq 2.$$

For such $n$ we have

$$\delta \leq 8\exp\{-2\log n\} = \frac{8}{n^2}$$

which is summable.

To show (S6), let $w^*$ be the true mixing weights from (2). For $i = 1, \ldots, n$ let $e_i$ be the $m \in [M]$ such that $X_i = (x_{i,1}, x_{i,2}) \overset{i.i.d.}{\sim} p_m^*$. Define

$$n_m = |\{i : e_i = m\}|, \qquad\qquad m = 1, \ldots, M$$

$$\alpha_{m,i,1}^* = \alpha_{m,i,2}^* = \begin{cases} \frac{1}{2n_m}, & e_i = m \\ 0, & \text{otherwise} \end{cases}, \qquad m = 1, \ldots, M$$

With this "oracle" assignment of weights, $p(x; \alpha_m^*)$ is just the regular KDE for $p_m^*$. Therefore, we may apply known results for consistency of standard KDEs. In particular, we will apply Theorem 3.1 of [3] which implies

$$\|p(\,\cdot\,; \alpha_m^*) - p_m^*\|_2 \overset{a.s.}{\longrightarrow} 0 \text{ as } n_m \to \infty \qquad (S7)$$

provided $k \in L^2$ and $\sum_n \frac{1}{n^2\sigma_n^d} < \infty$. Both of these conditions are satisfied by assumption in our setting. Furthermore, as $n \to \infty$ we have $\frac{n_m}{n} \to w_m^*$ almost surely, and therefore $n_m \to \infty$ almost surely.

Finally, we have

$$\inf_{\substack{w \in \Delta^M \\ \alpha \in \Delta_M^{2n}}} \|q - q_{w,\alpha}\|_2 \leq \|q - q_{w^*,\alpha^*}\|_2$$

$$= \left\|\sum_{m=1}^M w_m^*(p_m^* \times p_m^* - p(\,\cdot\,; \alpha_m^*) \times p(\,\cdot\,; \alpha_m^*))\right\|_2$$

$$\leq \sum_{m=1}^M w_m^*\|p_m^* \times p_m^* - p(\,\cdot\,; \alpha_m^*) \times p(\,\cdot\,; \alpha_m^*)\|_2$$

$$\leq \sum_{m=1}^M w_m^*(\|p_m^*\|_2 + \|p(\,\cdot\,; \alpha_m^*)\|_2)\|p_m^* - p(\,\cdot\,; \alpha_m^*)\|_2$$

$$\leq \sum_{m=1}^M w_m^* 3\|p_m^*\|_2\|p_m^* - p(\,\cdot\,; \alpha_m^*)\|_2$$

$$\overset{a.s.}{\longrightarrow} 0 \text{ as } n \to \infty,$$

where the fourth step uses Lemma 1 and the fifth step holds for $n$ sufficiently large (a.s.). This completes the proof. $\qquad\square$

## S4.3 Proof of Theorem 2: Arbitrary Group Size

We consider the problem for arbitrary group size as described in Section S3.2.1 of this document. The proof of Theorem 2 for arbitrary group size is similar to the proof for groups of size two. The main difference will be in use of Theorem 1a rather than Theorem 1 to invoke the Borel-Cantelli lemma.

254 **Theorem 2a.** *Given grouped observations of size $N \in \mathbb{Z}^+$, if $\sigma \to 0$ and $\frac{n\sigma^{2Nd}}{\log n} \to \infty$ as $n \to \infty$*
255 *then $\|q - \hat{q}\|_1 \xrightarrow{a.s.} 0$ as $n \to \infty$.*

256 *Proof.* We will appeal to Lemma 3.1 of [3] as we did for groups of size two. Namely, if $\int \hat{q} = 1$ and
257 $\|\hat{q} - q\|_2 \xrightarrow{a.s.} 0$, then $\|\hat{q} - q\|_1 \xrightarrow{a.s.} 0$. Our strategy again is to show $\|\hat{q} - q\|_2 \xrightarrow{a.s.} 0$. To do this it
258 suffices to show that

$$\|q - \hat{q}\|_2^2 - \inf_{\substack{w \in \Delta^M \\ \alpha \in \Delta_M^{Nn}}} \|q - q_{w,\alpha}\|_2^2 \xrightarrow{a.s.} 0 \tag{S8}$$

259 and

$$\inf_{\substack{w \in \Delta^M \\ \alpha \in \Delta_M^{Nn}}} \|q - q_{w,\alpha}\|_2 \xrightarrow{a.s.} 0. \tag{S9}$$

260 To show (S8), by the Borel-Cantelli lemma, it suffices to show that for all $\epsilon > 0$,

$$\sum_{n=1}^{\infty} P_q \left( \|q - \hat{q}\|_2^2 - \inf_{\substack{w \in \Delta^M \\ \alpha \in \Delta_M^{Nn}}} \|q - q_{w,\alpha}\|_2^2 \geq \epsilon \right) < \infty.$$

261 Thus let $\epsilon > 0$. By Theorem 1a, the probability in question is at most

$$\delta = 2(Nn)^N \exp\{-\frac{\sigma^{2Nd}(n-N)\epsilon^2}{8C_k^{2N}}\}$$

$$= 2N^N \exp\left\{ N \log n + \left( \frac{(n-N)\sigma^{2Nd}\epsilon^2}{8C_k^{2N}} \right) \right\}$$

$$= 2N^N \exp\left\{ -N \log n \left( \frac{(n-N)\sigma^{2Nd}\epsilon^2}{8C_k^{2N}N\log n} - 1 \right) \right\}.$$

262 By assumption on the growth of $n$ and $\sigma$, there exists $N_\epsilon$ such that for all $n \geq N_\epsilon$,

$$\frac{(n-N)\sigma^{2Nd}\epsilon^2}{8C_k^{2N}N\log n} \geq 2.$$

263 For such $n$ we have

$$\delta \leq 2N^N \exp\{-N\log n\} = \frac{2N^N}{n^N}$$

264 which is summable for $N > 1$.

265 To show (S9), let $w^*$ be the true mixing weights from (S4). For $i = 1, \ldots, n$ let $e_i$ be the $m \in [M]$
266 such that $X_i = (x_{i,1}, x_{i,2}, \ldots, x_{i,N}) \overset{i.i.d.}{\sim} p_m^*$. Define

$$n_m = |\{i : e_i = m\}|, \qquad m = 1, \ldots, M$$

$$\alpha_{m,i,j}^* = \begin{cases} \frac{1}{n_m N}, & e_i = m \\ 0, & \text{otherwise} \end{cases}, \qquad m = 1, \ldots, M, \ j = 1, \ldots, N$$

267 We are again using an "oracle" assignment of weights, so $p(x; \alpha_m^*)$ is just the regular KDE for $p_m^*$.
268 Therefore, we may again apply Theorem 3.1 of [3] which implies

$$\|p(\cdot\,; \alpha_m^*) - p_m^*\|_2 \xrightarrow{a.s.} 0 \text{ as } n_m \to \infty \tag{S10}$$

269 provided $k \in L^2$ and $\sum_n \frac{1}{n^2 \sigma_n^d} < \infty$. Both of these conditions are satisfied by assumption in our
270 setting. Furthermore, as $n \to \infty$ we have $\frac{n_m}{n} \to w_m^*$ almost surely, and therefore $n_m \to \infty$ almost
271 surely.

Finally, we have

$$\inf_{\substack{w \in \Delta^M \\ \alpha \in \Delta_M^{Nn}}} \|q - q_{w,\alpha}\|_2 \leq \|q - q_{w^*,\alpha^*}\|_2$$

$$= \left\| \sum_{m=1}^M w_m^* (p_m^{* \times N} - p(\cdot\,; \alpha_m^*)^{\times N}) \right\|_2$$

$$\leq \sum_{m=1}^M w_m^* \| p_m^{* \times N} - p(\cdot\,; \alpha_m^*)^{\times N} \|_2$$

$$\leq \sum_{m=1}^M w_m^* \left( \sum_{b=1}^N \|p_m^*\|_2^{N-b} \|p(\cdot\,; \alpha_m^*)\|_2^{b-1} \right) \|p_m^* - p(\cdot\,; \alpha_m^*)\|_2$$

$$\leq \sum_{m=1}^M w_m^* \left( \sum_{b=1}^N 2^{b-1} \|p_m^*\|_2^{N-1} \right) \|p_m^* - p(\cdot\,; \alpha_m^*)\|_2$$

$$\xrightarrow{a.s.} 0 \text{ as } n \to \infty,$$

where the penultimate step uses (S10) and Corollary 1, and the final step holds for $n$ sufficiently large
(a.s.). This completes the proof. $\qquad \square$

## S5   Background on the Grouped Sample Setting and Proof of Theorem 3

Here we prove Theorem 3. We will be proving a general and more technical version of this theorem,
Theorem 5, from which Theorem 3 is a direct consequence. First we will introduce some background
to the problem setting which was introduced in [12]. **This section uses its own notation which does
not extend to other parts of the supplement or main text**.

### S5.1   Identifiability in the Grouped Sample Setting

We will be concerned with probability measures on a measurable space $(\Omega, \mathcal{F})$. Let $\delta$ be the Dirac
measure. Let $\mathcal{D}$ be the set of probability measures on $(\Omega, \mathcal{F})$. We call a probability measure on $\mathcal{D}$ of
the form

$$\mathscr{P} = \sum_{i=1}^m a_i \delta_{\mu_i}$$

a *mixture of measures* [12]. For all mixtures of measures we will assume that $a_i > 0$ for all $i$ and
$\mu_i \neq \mu_j$ when $i \neq j$ so that $m$ is the number of distinct mixture components. The *grouped sample*
setting from [12] considers the situation where samples come in groups of size $n$ by first sampling
a random measure component from a mixture of measures $\gamma \sim \mathscr{P}$, which is then sampled iid $n$
times. So one has access to samples of the form $\mathbf{X} = (X_1, \ldots, X_n)$ with $X_1, \ldots, X_n \overset{iid}{\sim} \gamma$. In this
situation the identifiability of $\mathscr{P}$ depends on whether the distribution of $\mathbf{X}$ is uniquely determined by
$\mathscr{P}$ and the number of samples per group $n$. To this end [12] introduced the $V_n$ operator which maps
a mixture of measures to the distribution of $\mathbf{X}$:

$$V_n \left( \sum_{i=1}^m a_i \delta_{\mu_i} \right) = \sum_{i=1}^m a_i \mu_i^{\times n},$$

where $\mu^{\times n}$ denotes the product measure $n$ times. We note that $n = 1$ corresponds to a typical
mixture model where each mixture component is sampled once after being selected and there is no
grouped sample structure. For the grouped sample setting [12] introduces the following notion of
identifiability.

**Definition 1.** *A mixture of measures,* $\mathscr{P} = \sum_{i=1}^m a_i \delta_{\mu_i}$, *is called $n$-identifiable if there does not
exist a different mixture of measures* $\mathscr{Q} = \sum_{j=1}^{m'} b_j \delta_{\nu_j}$, *with* $m' \leq m$, *such that* $V_n(\mathscr{P}) = V_n(\mathscr{Q})$.

A *completely* rigorous mathematical treatment of the previous notions is a bit involved and can be
found in [12]. In [12] it is shown that if the mixture components are jointly irreducible then a mixture
of measures is 2-identifiable, if they are linearly independent then they are 3-identifiable, and that any
mixture of measures with $m$ components is $(2m - 1)$-identifiable.

## S5.2  Notation

Before we state and prove the main theorem of this section we need to first introduce some notation.

Let $S_m$ be the symmetric group over $m$ symbols. Abusing notation slightly we will let the elements of $S_m$ be a group action on $[m]$ as well as $\mathbb{R}^m$. On $\mathbb{R}^m$ it is defined as the following

$$\sigma\left([x_1,\ldots,x_m]^T\right) = \left[x_{\sigma(1)},\ldots,x_{\sigma(m)}\right]^T. \tag{S11}$$

We also let $S_m$ be an operator where $S_m \cdot x$ is the orbit of $x$, i.e.

$$S_m \cdot x \triangleq \{\sigma(x) : \sigma \in S_m\}. \tag{S12}$$

Recall that for a pair of Hilbert spaces $H, H'$ the direct sum $H \oplus H'$ is a Hilbert space with elements of the form $x \oplus x'$ and inner product defined as $\langle x \oplus x', y \oplus y'\rangle = \langle x, y\rangle + \langle x', y'\rangle$. For a pair of Banach spaces $B, B'$ we define the direct sum via the norm $\|b \oplus b'\|_{B \oplus B'} \triangleq \|b\|_B + \|b'\|_{B'}$ which is itself a Banach space ([1] p. 183).

For a pair of Hilbert spaces $H, H'$ let $H \otimes H'$ be the tensor product of these two spaces and $h \otimes h'$ be the tensor product of vectors $h \in H$ and $h' \in H'$. For a vector in a Hilbert space $h$ let $h^{\otimes n}$ denote the tensor power, i.e. $\underbrace{h \otimes \cdots \otimes h}_{n \text{ times}}$.

In the following the space of finite signed measures is equipped with the total variation topology and unadorned norms refer to the total variation norm on finite signed measures, which forms a Banach space. Norms for various Lebesgue spaces will have the associated subscript. Finally we note that for two Hilbert spaces of square-integrable functions over $\sigma$-finite measure spaces $L^2(\Omega, \mathcal{F}, \mu)$ and $L^2(\Omega', \mathcal{F}', \mu')$ we have that $L^2(\Omega, \mathcal{F}, \mu) \otimes L^2(\Omega', \mathcal{F}', \mu') \cong L^2(\Omega \times \Omega', \mathcal{F} \times \mathcal{F}', \mu \times \mu')$ via an isomorphism $f \otimes f' \mapsto f \times f'$ ([4] Example 2.6.11) and we will use a 2 subscript for both norms.

## S5.3  Full Theorem Statement and Proof

The following is the full general version of Theorem 3 and the main result of this section.

**Theorem 5.** *Let $(\Omega, \mathcal{F})$ be a measurable space, $\mathscr{P} = \sum_{j=1}^m a_j \delta_{\mu_j}$ a mixture of measures on that space which is $n$-identifiable, and $\mathscr{P}_i = \sum_{j=1}^{m_i'} b_{i,j}\delta_{\nu_{i,j}}$ a sequence of mixtures of measures with $m_i' \leq m$ for all $i$, such that $V_n(\mathscr{P}_i) \to V_n(\mathscr{P})$. Then $m_i' \to m$ and there exists a sequence of permutations $\sigma_i$ such that $\sigma_i(b_i) \to a$ and $\nu_{i,\sigma_i(j)} \to \mu_j$ for all $j$.*

Essentially this says that as one finds grouped sample distributions $V_n(\mathscr{Q}_i)$ which approach the true grouped sample distribution $V_n(\mathscr{P})$ the mixture of measures $\mathscr{Q}_i$ will automatically recover the true mixing weights and components from $\mathscr{P}$ so long as $\mathscr{P}$ is $n$-identifiable. In other words, one simply needs to fit the grouped distribution $V_n(\mathscr{P})$ well to get a good estimate of the mixture components. Theorem 3 from the main text is a direct consequence of Theorem 5.

**Corollary 2** (Theorem 3). *Let $\sum_{m=1}^M w_m p_m$ be an $N$-identifiable mixture model, and $\sum_{m=1}^M \hat{w}_{m,j}\hat{p}_{m,j}$ be a sequence of mixture models such that $\left\|\sum_{m=1}^M \hat{w}_{m,j}\hat{p}_{m,j}^{\times N} - \sum_{m=1}^M w_m p_m^{\times N}\right\|_1 \to 0$. Then there is a sequence of permutations $\sigma_j$ so that $\hat{w}_{\sigma_j(m),j} \to w_m$ and $\left\|\hat{p}_{\sigma_j(m),j} - p_m\right\|_1 \to 0$ for all $m$.*

We introduce some preliminary results before proving Theorem 5. The following lemma will be needed for our proof.

**Lemma 2.** *Let $\mathscr{P}$ and $\mathscr{Q}$ be mixtures of measures, then $\|V_{n'}(\mathscr{P}) - V_{n'}(\mathscr{Q})\| \leq \|V_n(\mathscr{P}) - V_n(\mathscr{Q})\|$ for all $n' \leq n$.*

*Proof of Lemma 2.* From [2] (Section 3.1 Exercise 7a) we have the following

$$\|V_n(\mathscr{P}) - V_n(\mathscr{Q})\|$$

$$= \sup\left\{ \sum_{i=1}^{k} |(V_n(\mathscr{P}) - V_n(\mathscr{Q}))(E_i)| : \right.$$

$$\left. k \in \mathbb{N}, E_1, \dots, E_k \in \mathcal{F}^{\times n} \text{ are disjoint, and } \bigcup_{i=1}^{k} E_i = \Omega^{\times n} \right\}$$

$$\geq \sup\left\{ \sum_{i=1}^{k} \left|(V_n(\mathscr{P}) - V_n(\mathscr{Q}))(E_i \times \Omega^{\times n - n'})\right| : \right.$$

$$\left. k \in \mathbb{N}, E_1, \dots, E_k \in \mathcal{F}^{\times n'} \text{ are disjoint, and } \bigcup_{i=1}^{k} E_i = \Omega^{\times n'} \right\}$$

$$= \sup\left\{ \sum_{i=1}^{k} |(V_{n'}(\mathscr{P}) - V_{n'}(\mathscr{Q}))(E_i)| : \right.$$

$$\left. k \in \mathbb{N}, E_1, \dots, E_k \in \mathcal{F}^{\times n'} \text{ are disjoint, and } \bigcup_{i=1}^{k} E_i = \Omega^{\times n'} \right\}$$

$$= \|V_{n'}(\mathscr{P}) - V_{n'}(\mathscr{Q})\|.$$

$\square$

The following lemma is the main workhorse in the proof of Theorem 5.

**Lemma 3.** *Let* $(\Omega, \mathcal{F})$ *be a measurable space,* $\mathscr{P} = \sum_{i=1}^{m} a_i \delta_{\mu_i}$ *a mixture of measures on that space,* $n \in \mathbb{N}$, *and* $\mathscr{P}_i = \sum_{j=1}^{m'} b_j \delta_{\nu_{i,j}}$ *a sequence of mixtures of measures ($m'$ is fixed) with such that* $V_n(\mathscr{P}_i) \to V_n(\mathscr{P})$ *(b does not depend on i). Then there exists a subsequence* $i_k$ *and a collection of probability measures* $\nu_1, \dots, \nu_{m'}$ *such that* $\nu_{i_k,j} \to \nu_j$ *for all j and* $V_n(\mathscr{P}) = V_n\left(\sum_{j=1}^{m'} b_j \delta_{\nu_j}\right)$.

*Proof of Lemma 3.* We will use bold symbols to represent elements that depend on $i$, e.g. $\boldsymbol{\nu}_j = \nu_{i,j}$. Let $\bar{\mu} = \sum_{k=1}^{m} a_k \mu_k$. By the Lebesgue-Radon-Nikodym Theorem ([2] Theorem 3.8) there exists series of measures $\boldsymbol{\lambda}_1, \dots, \boldsymbol{\lambda}_{m'}$ and $\boldsymbol{\rho}_1, \dots, \boldsymbol{\rho}_{m'}$ such that $\boldsymbol{\nu}_k = \boldsymbol{\lambda}_k + \boldsymbol{\rho}_k$ with $\boldsymbol{\lambda}_k \perp \bar{\mu}$ and $\boldsymbol{\rho}_k \ll \bar{\mu}$ for all $k \in [m']$.

For some fixed $\ell$ let $\mathbf{A}_\ell$ be the sequence of measurable sets such that $\boldsymbol{\lambda}_\ell\left(\cdot \cap \mathbf{A}_\ell\right) = \boldsymbol{\lambda}_\ell$ and $\bar{\mu}\left(\mathbf{A}_\ell\right) = 0$, this is possible since $\boldsymbol{\lambda}_\ell \perp \bar{\mu}$. From Lemma 2 we have that

$$\left\|\sum_{k=1}^{m} a_k \mu_k - \sum_{j=1}^{m'} b_j \boldsymbol{\nu}_j\right\| \to 0 \Rightarrow \left|\sum_{k=1}^{m} a_k \mu_k(\mathbf{A}_\ell) - \sum_{j=1}^{m'} b_j \boldsymbol{\nu}_j(\mathbf{A}_\ell)\right| \to 0 \tag{S13}$$

$$\Rightarrow \left|b_\ell \boldsymbol{\rho}_\ell(\mathbf{A}_\ell) + b_\ell \boldsymbol{\lambda}_\ell(\mathbf{A}_\ell) + \sum_{j \in [m'] \setminus \{\ell\}} b_j \boldsymbol{\nu}_j(\mathbf{A}_\ell)\right| \to 0 \tag{S14}$$

$$\Rightarrow \left|b_\ell \boldsymbol{\lambda}_\ell(\mathbf{A}_\ell) + \sum_{j \in [m'] \setminus \{\ell\}} b_j \boldsymbol{\nu}_j(\mathbf{A}_\ell)\right| \to 0. \tag{S15}$$

Because all of the summands inside the absolute value on the last line are positive we have that $\|\boldsymbol{\lambda}_\ell\| \to 0$ and thus $\|\boldsymbol{\rho}_\ell\| \to 1$. Eventually in our sequence we must have that $\|\boldsymbol{\rho}_\ell\| > 0$, so eventualy in our subsequence we can define $\boldsymbol{\nu}'_\ell = \boldsymbol{\rho}_\ell / \|\boldsymbol{\rho}_\ell\|$ which is now a sequence of probability measures which are absolutely continuous with respect to $\bar{\mu}$ and $\|\boldsymbol{\nu}'_\ell - \boldsymbol{\nu}_\ell\| \to 0$.

From this we have that there exists sequences of probability measures $\boldsymbol{\nu}'_1, \dots, \boldsymbol{\nu}'_{m'}$ such that $\|\boldsymbol{\nu}_k - \boldsymbol{\nu}'_k\| \to 0$ and $\boldsymbol{\nu}'_k \ll \bar{\mu}$ for all $k \in [m']$. Lemma 3.3.7 in [11] states that, for probablity measures over the same domain $\xi_1, \dots \xi_d, \gamma_1, \dots, \gamma_d$ that $\left\|\prod_{j=1}^{d} \xi_j - \prod_{k=1}^{d} \gamma_k\right\| \leq \sum_{k=1}^{d} \|\xi_k - \gamma_k\|$.

359    It follows therefore that $\left\| {\boldsymbol{\nu}'}_k^{\times n} - \boldsymbol{\nu}_k^{\times n} \right\| \to 0$ for all $k$ and

$$\left\| \sum_{k=1}^{m} a_k \mu_k^{\times n} - \sum_{j=1}^{m'} b_j {\boldsymbol{\nu}'}_j^{\times n} \right\| \to 0. \tag{S16}$$

360    For some fixed $\ell$ let $\mathbf{q}'_\ell$ be the Radon-Nikodym derivative of $\boldsymbol{\nu}'_\ell$ with respect to $\bar{\mu}$. Let $\mathbf{B}_\ell = $
361    ${\mathbf{q}'}_\ell^{-1} \left([2/b_\ell, \infty)\right)$. We have the following

$$\sum_{k=1}^{m'} b_k \boldsymbol{\nu}'_k(\mathbf{B}_\ell) \geq b_\ell \boldsymbol{\nu}'_\ell(\mathbf{B}_\ell) \tag{S17}$$

$$\geq b_\ell \int_{\mathbf{B}_1} 2/b_\ell d\bar{\mu} \tag{S18}$$

$$\geq 2\bar{\mu}(\mathbf{B}_\ell). \tag{S19}$$

362    From Lemma 2 applied to (S16) we have that $\left| \sum_{k=1}^{m'} b_k \boldsymbol{\nu}'_k(\mathbf{B}_\ell) - \bar{\mu}(\mathbf{B}_\ell) \right| \to 0$ and be-
363    cause $\left| \sum_{k=1}^{m'} b_k \boldsymbol{\nu}'_k(\mathbf{B}_\ell) - \bar{\mu}(\mathbf{B}_\ell) \right| \geq \bar{\mu}(\mathbf{B}_\ell)$ it follows that $\bar{\mu}(\mathbf{B}_\ell) \to 0$. Now we have that
364    $\sum_{k=1}^{m'} b_k \boldsymbol{\nu}'_k(\mathbf{B}_\ell) \to 0$ and thus $\boldsymbol{\nu}'_\ell(\mathbf{B}_\ell) \to 0$.

365    Because $\boldsymbol{\nu}'_\ell \left( {\mathbf{q}'}_\ell^{-1} \left([2/b_\ell, \infty)\right) \right) \to 0$ and therefore $\boldsymbol{\nu}'_\ell \left(\mathbf{B}_\ell^C\right) \to 1$, for sufficiently large $i$ we can now
366    define a sequence of probability measures $\boldsymbol{\nu}''_\ell$ via $\boldsymbol{\nu}''_\ell(A) = \boldsymbol{\nu}'_\ell(A \cap \mathbf{B}_\ell^C)/\boldsymbol{\nu}'_\ell(\mathbf{B}_\ell^C)$. We have that

$$\|\boldsymbol{\nu}'_\ell - \boldsymbol{\nu}''_\ell\| = \left\| \left( \boldsymbol{\nu}'_\ell \left(\mathbf{B}_\ell \cap \cdot\right) + \boldsymbol{\nu}'_\ell \left(\mathbf{B}_\ell^C \cap \cdot\right) \right) - \left( \boldsymbol{\nu}''_\ell \left(\mathbf{B}_\ell \cap \cdot\right) + \boldsymbol{\nu}''_\ell \left(\mathbf{B}_\ell^C \cap \cdot\right) \right) \right\| \tag{S20}$$

$$\leq \left\| \boldsymbol{\nu}'_\ell \left(\mathbf{B}_\ell \cap \cdot\right) - \boldsymbol{\nu}''_\ell \left(\mathbf{B}_\ell \cap \cdot\right) \right\| + \left\| \boldsymbol{\nu}'_\ell \left(\mathbf{B}_\ell^C \cap \cdot\right) - \boldsymbol{\nu}''_\ell \left(\mathbf{B}_\ell^C \cap \cdot\right) \right\| \tag{S21}$$

$$= \boldsymbol{\nu}'_\ell \left(\mathbf{B}_\ell\right) + \left\| \boldsymbol{\nu}'_\ell \left(\mathbf{B}_\ell^C \cap \cdot\right) - \boldsymbol{\nu}'_\ell \left(\mathbf{B}_\ell^C \cap \cdot\right) / \boldsymbol{\nu}'_\ell \left(\mathbf{B}_\ell^C\right) \right\| \tag{S22}$$

$$= \boldsymbol{\nu}'_\ell \left(\mathbf{B}_\ell\right) + \left| 1 - 1/\boldsymbol{\nu}'_\ell \left(\mathbf{B}_\ell^C\right) \right| \left\| \boldsymbol{\nu}'_\ell \left(\mathbf{B}_\ell \cap \cdot\right) \right\| \tag{S23}$$

367    which goes to zero, so $\|\boldsymbol{\nu}''_\ell - \boldsymbol{\nu}_\ell\| \to 0$. Note that $\boldsymbol{\nu}''_\ell$ is a sequence of probability measures with
368    Radon-Nikodym derivatives $\mathbf{q}''_\ell \triangleq \mathbf{q}'_\ell \mathbf{1}_{\mathbf{B}_\ell^C} / \boldsymbol{\nu}'_\ell \left(\mathbf{B}_\ell^C\right)$ ($\mathbf{1}$ is the indicator function) and thus

$$\sup_x \mathbf{q}''_\ell(x) = \sup_x \mathbf{q}'_\ell(x) \mathbf{1}_{\mathbf{B}_\ell^C}(x) / \boldsymbol{\nu}_\ell \left(\mathbf{B}_\ell^C\right) \leq 2/(b_\ell \boldsymbol{\nu}_\ell \left(\mathbf{B}_\ell^C\right))$$

and since $\boldsymbol{\nu}_\ell \left(\mathbf{B}_\ell^C\right) \to 1$ eventually $\|\mathbf{q}''_\ell\|_\infty \leq 3/b_\ell$. From this we have that $\mathbf{q}''_\ell \in L^1(\Omega, \mathcal{F}, \bar{\mu}) \cap$
$L^\infty(\Omega, \mathcal{F}, \bar{\mu})$ and $\|\mathbf{q}''_\ell\|_\infty$ is a bounded sequence. From Hölders's Inequality we have that

$$= \|\mathbf{q}''_\ell\|_2^2 = \|\mathbf{q}''_\ell \mathbf{q}''_\ell\|_1^2 \leq \|\mathbf{q}''_\ell\|_1 \|\mathbf{q}''_\ell\|_\infty = \|\mathbf{q}''_\ell\|_\infty$$

369    so $\mathbf{q}''_\ell$ is a bounded sequence in $L^2(\Omega, \mathcal{F}, \bar{\mu})$.

370    We now define $\boldsymbol{\nu}''_1, \ldots, \boldsymbol{\nu}''_{m'}$ $\mathbf{q}''_1, \ldots, \mathbf{q}''_{m'}$ similarly. There exists $\beta$ such that $\|\mathbf{q}''_j\|_\infty \leq \beta$ and
371    $\|\mathbf{q}''_j\|_2 \leq \beta$ along the whole series and for all $j$. Let $p_1, \ldots, p_m$ be the radon Nikodym derivatives
372    for $\mu_1, \ldots, \mu_m$ with respect to $\bar{\mu}$, again these are in $L^1(\Omega, \mathcal{F}, \bar{\mu}) \cap L^2(\Omega, \mathcal{F}, \bar{\mu}) \cap L^\infty(\Omega, \mathcal{F}, \bar{\mu})$. To
373    see this note that $p_i \leq 1/a_i$ otherwise we have that

$$\mu_i \left( p_i^{-1} \left((1/a_i, \infty)\right) \right) = \int_{p_i^{-1}((1/a_i, \infty))} p_i d\bar{\mu}$$

$$> \int_{p_i^{-1}((1/a_i, \infty))} 1/a_i d\bar{\mu}$$

$$> \sum_j \int_{p_i^{-1}((1/a_i, \infty))} 1/a_i a_j d\mu_j$$

$$\geq \mu_i \left( p_i^{-1} \left((1/a_i, \infty)\right) \right)$$

374 a contradiction. Now we have

$$\left\| \sum_{k=1}^{m} a_k p_k^{\times n} - \sum_{j=1}^{m'} b_j \mathbf{q}_j''^{\times n} \right\|_1 \to 0. \tag{S24}$$

375 and Lemma 2 implies

$$\left\| \sum_{k=1}^{m} a_k p_k^{\times 2} - \sum_{j=1}^{m'} b_j \mathbf{q}_j''^{\times 2} \right\|_1 \to 0. \tag{S25}$$

376 From Hölder's Inequality ($\|f\|_2^2 \le \|f\|_1 \|f\|_\infty$) we have that

$$\left\| \sum_{k=1}^{m} a_k p_k^{\times 2} - \sum_{j=1}^{m'} b_j \mathbf{q}_j''^{\times 2} \right\|_2 \to 0 \tag{S26}$$

377 and

$$\left\| \sum_{k=1}^{m} a_k p_k^{\otimes 2} - \sum_{j=1}^{m'} b_j \mathbf{q}_j''^{\otimes 2} \right\|_2 \to 0. \tag{S27}$$

378 Let $S = \operatorname{span}(\{p_1, \ldots, p_m\})$ and $\ell \in [m']$ be arbitrary. We have that $\mathbf{q}_\ell'' = \operatorname{proj}_S(\mathbf{q}_\ell'') +$
379 $\operatorname{proj}_{S^\perp}(\mathbf{q}_\ell'')$, noting that the summands in the decomposition are both $L^2$ bounded sequences. So
380 now we have that

$$\left\langle \sum_{k=1}^{m'} b_k \mathbf{q}_k''^{\otimes 2} - \sum_{j=1}^{m} a_j p_j^{\otimes 2}, \operatorname{proj}_{S^\perp}(\mathbf{q}_\ell'')^{\otimes 2} \right\rangle \to 0 \tag{S28}$$

$$\Rightarrow \left\langle \sum_{j=1}^{m'} b_j \mathbf{q}_j''^{\otimes 2}, \operatorname{proj}_{S^\perp}(\mathbf{q}_\ell'')^{\otimes 2} \right\rangle \to 0 \tag{S29}$$

$$\Rightarrow b_\ell \left\langle \operatorname{proj}_{S^\perp}(\mathbf{q}_\ell'')^{\otimes 2}, \operatorname{proj}_{S^\perp}(\mathbf{q}_\ell'')^{\otimes 2} \right\rangle + \sum_{j \in [m'] \setminus \{\ell\}} b_j \left\langle \mathbf{q}_j'', \operatorname{proj}_{S^\perp}(\mathbf{q}_j'') \right\rangle^2 \to 0 \tag{S30}$$

$$\Rightarrow b_\ell \left\| \operatorname{proj}_{S^\perp}(\mathbf{q}''_\ell) \right\|_2^4 \to 0. \tag{S31}$$

381 From this we have that $\|\operatorname{proj}_S(\mathbf{q}_k'') - \mathbf{q}_k''\|_2 \to 0$ for all $k$. Since $\bigoplus_{j=1}^{m'} \operatorname{proj}_S(\mathbf{q}_j'')$ is a $L^2$ bounded
382 sequence on a finite dimensional space by the Bolzano-Weierstrass theorem it has a convergent
383 subsequence which converges to $\bigoplus_{j=1}^{m'} q_j''$ so $\mathbf{q}_j'' \to q_j''$ in $L^2$. From Hölder's Inequality we have
384 that, along this subsequence

$$\|\mathbf{q}_k'' - q_k''\|_1 \le \|\mathbf{q}_k'' - q_k''\|_2 \|1\|_2 \le \|\mathbf{q}_k'' - q_k''\|_2 \sqrt{\int 1^2 d\bar{\mu}} = \|\mathbf{q}_k'' - q_k''\|_2 \to 0 \tag{S32}$$

385 so $q_k''$ is a probability density for all $k$, since they must be nonnegative to converge and integrate to
386 one. Now we have that

$$\sum_{j=1}^{m} a_j p_j^{\times n} = \sum_{k=1}^{m'} b_k q_k''^{\times n}. \tag{S33}$$

387 And defining $\nu_k$ as the probability measure associated with $q_k''$ we have that there exists a subsequence
388 such that $\|\boldsymbol{\nu}_k - \nu_k\| \to 0$ for all $k$ and

$$\sum_{j=1}^{m} a_j \mu_j^{\times n} = \sum_{k=1}^{m'} b_k \nu_k^{\times n}. \tag{S34}$$

389 $\qquad\qquad\qquad\qquad\qquad\qquad\qquad\qquad\qquad\qquad\qquad\qquad\qquad\qquad\qquad\qquad\qquad\qquad \square$

390    We can now prove Theorem 5.

391    *Proof of Theorem 5.* To help lighten notation we will simply bold some elements which depend
392    on the sequence $\mathscr{P}_i$. Let $\mathscr{P}_i = \sum_{j=1}^{m'_i} \mathbf{b}_j \delta_{\boldsymbol{\nu}_j}$ be a sequence of mixtures of measures ($\mathbf{b}_j$, $\boldsymbol{\nu}_j$ are
393    functions of $i$) such that $V_n(\mathscr{P}_i) \to V_n(\mathscr{P})$.

394    We define $\widetilde{\mathbf{b}}$ a sequence in $\Delta^m$ so that $\widetilde{\mathbf{b}}_j = \mathbf{b}_j$ for $j \leq m'_i$ and $\widetilde{\mathbf{b}}_k = 0$ for $k > m'_i$. Consider the
395    case where there exists no sequence of permutations such that $\boldsymbol{\sigma}(\widetilde{\mathbf{b}}) \to a$. From this it would follow
396    that there exists a subsequence on $i$ and $\varepsilon > 0$ such that $\left\| \widetilde{\mathbf{b}} - \sigma(a) \right\| > \varepsilon$ for all $\sigma \in S_m$. The space

$$\Delta^m \cap \left( \bigcap_{\sigma \in S_m} \text{ball}\,(\sigma(a), \varepsilon)^C \right) \tag{S35}$$

397    is compact (the ball is open) so there exists a sub-subsequence of $i$ where $\widetilde{\mathbf{b}}$ converges to a point
398    $b \notin S_m \cdot a$. Let $I \subset [m]$ be the indices of $b$ which are nonzero and $m' = \max(I)$. For sufficiently
399    large $i$ along our sub-subsequence we have that $m'_i \geq m'$ and furthermore

$$\left\| \sum_{j=1}^{m'_i} \mathbf{b}_j \boldsymbol{\nu}_j^{\times n} - \sum_{k \in I} b_k \boldsymbol{\nu}_k^{\times n} \right\| \leq \left\| \sum_{j \in I} \mathbf{b}_j \boldsymbol{\nu}_j^{\times n} - \sum_{k \in I} b_k \boldsymbol{\nu}_k^{\times n} \right\| + \left\| \sum_{j \in I^C} \mathbf{b}_j \boldsymbol{\nu}_j^{\times n} \right\| \tag{S36}$$

$$\leq \left\| \sum_{j \in I} (\mathbf{b}_j - b_j) \boldsymbol{\nu}_j^{\times n} \right\| + \left| \sum_{j \in I^C} \mathbf{b}_j \right| \tag{S37}$$

$$\leq \sum_{j \in I} |(\mathbf{b}_j - b_j)| + \left| \sum_{j \in I^C} \mathbf{b}_j \right| \to 0 \tag{S38}$$

400    and therefore

$$\left\| \sum_{k=1}^{m} a_k \mu_k^{\times n} - \sum_{j \in I} b_j \boldsymbol{\nu}_j^{\times n} \right\| \to 0. \tag{S39}$$

401    From Lemma 3 we have that there exists a subsequence of this sub-subsequence such that for $k \in I$
402    there exists probability measures $\nu_k$ with $\|\nu_k - \boldsymbol{\nu}_k\| \to 0$ and

$$\sum_{k=1}^{m} a_k \mu_k^{\times n} = \sum_{j \in I} b_j \nu_j^{\times n}. \tag{S40}$$

403    If $|I| < m$ or $\nu_j = \nu_k$ for any $k \neq j$ and $j, k \in I$ then we have clearly violated identifiability since
404    we can construct a mixture of measures $\mathscr{P}'$ with fewer components than $\mathscr{P}$ and $V_n(\mathscr{P}') = V_n(\mathscr{P})$.
405    If $|I| = m$ (i.e. $I = [m]$) and $\nu_j$ are all distinct we have also arrived at a contradiction since letting
406    $\mathscr{P}' = \sum_{j=1}^{m} b_j \delta_{\nu_j} \neq \mathscr{P}$ because there exists no $\sigma$ such that $\sigma(b) = a$ and $V_n(\mathscr{P}') = V_n(\mathscr{P})$,
407    contradicting identifiability.

408    So we have that for sufficiently large $i$ that $m'_i = m$ and there exists at least one sequence $\boldsymbol{\sigma}$ such
409    that $\boldsymbol{\sigma}(\mathbf{b}) \to a$. So let $\left\| \sum_{k=1}^{m} a_k \mu_k^{\times n} - \sum_{j=1}^{m} \mathbf{b}_j \boldsymbol{\nu}_j^{\times n} \right\| \to 0$. From what we have just shown, we
410    can permute the indices and, without loss of generality, we can assume that $\mathbf{b} \to a$. So now we have
411    that $\left\| \sum_{i=1}^{m} a_i \mu_i^{\times n} - \sum_{j=1}^{m} a_j \boldsymbol{\nu}_j^{\times n} \right\| \to 0$.

Let $\widetilde{S}_m \subset S_m$ be the subgroup of permutations such that $\sigma(a) = a$ for $\sigma \in \widetilde{S}_m$ (also known as the
*stabilizer* of $a$). Note that if $a_1, \ldots, a_m$ are distinct then $\widetilde{S}_m$ only contains the identity. We proceed
by contradiction: suppose there exists no sequence of permutations $\boldsymbol{\sigma} \in \mathbb{N}^{\widetilde{S}_m}$ such that $\boldsymbol{\nu}_{\boldsymbol{\sigma}(k)} \to \mu_k$
for all $k$. From this it follows that there exists a subsequence and a $\varepsilon > 0$, such that $\bigoplus_{k=1}^{m} \boldsymbol{\nu}_k$
does not lie in $\bigcap_{\sigma \in \widetilde{S}_m} \left( \text{ball}\,\left( \bigoplus_{k=1}^{m} \mu_{\sigma(k)} \right), \varepsilon \right)^C$. From Lemma 3 there exists probability measures,

$\nu_1, \ldots, \nu_m$ such that for some subsequence $\|\boldsymbol{\nu}_k - \nu_k\| \to 0$ for all $k$ and

$$\sum_{j=1}^{m} a_j \mu_j^{\times n} = \sum_{k=1}^{m} a_k \nu_k^{\times n}.$$

Because $\bigcap_{\sigma \in \widetilde{S}_m} \left( \text{ball} \left( \bigoplus_{k=1}^{m} \mu_{\sigma(k)} \right), \varepsilon \right)^C$ is closed we have $\bigoplus_{j=1}^{m} \nu_j \in \bigcap_{\sigma \in \widetilde{S}_m} \left( \text{ball} \left( \bigoplus_{k=1}^{m} \mu_{\sigma(k)} \right), \varepsilon \right)^C$ and there exists no $\sigma \in \widetilde{S}_m$ such that $\nu_{\sigma(k)} = \mu_k$ for all $k$ so. Setting $\mathscr{P}' = \sum_{k=1}^{m} a_k \delta_{\nu_k}$ we have that $\mathscr{P}' \neq \mathscr{P}$ but $V_n(\mathscr{P}') = V_n(\mathscr{P})$, a contradiction.

$\square$

## S6    General Version of Corollary 1

Here we present the general version of Corollary 1 which guarantees recovery of the true mixture components using our estimator for any mixture model, provided there are a sufficient number of samples per group. For a mixture model $p = \sum_{m=1}^{M} w_m^* p_m^*$, using the estimator $\hat{q}$ from Section S3.2.1 to estimate (S4):

$$q(y_1, y_2, \ldots, y_N) = \sum_{m=1}^{M} w_m^* p_m^*(y_1) p_m^*(y_2) \ldots p_m^*(y_N), \quad y_1, y_2, \ldots, y_N \in \mathbb{R}^d.$$

combining Theorem 2a and Theorem 3 gives the following result.

**Corollary 3.** *If $\sigma \to 0$ and $\frac{n\sigma^{2Nd}}{\log n} \to \infty$ as $n \to \infty$, and $p$ is $N$-identifiable (e.g. $N = 2M - 1$), then $\hat{w}_m \overset{a.s.}{\to} w_m^*$ and $\|p(\cdot; \hat{\alpha}_m) - p_m^*\|_1 \overset{a.s.}{\to} 0$, up to a permutation.*

## Footnotes

[1] https://github.com/fivethirtyeight/russian-troll-tweets

[2] https://github.com/qiang2100/STTM