[Reviews · NeurIPS 2020]

Review 1

Summary and Contributions: This paper studies the problem of estimating nonparamatric mixture models from grouped observations (i.e. pairs of data (x_1, x_2) where x_1 and x_2 are sampled from the same mixture component). The authors provided the first consistency result for this problem using a KDE-type estimator. ---------------------------- Post-rebuttal update: I would like to thank the authors for their response. After reading the rebuttal and discussing with other reviewers, I've decided to raise my score to 6. I agree with R3 that the problem considered in this paper is important, and would love to see more work in this area in the future. It would be great if the authors can include a paragraph about the main technical barriers to finite-sample rate (just like in the rebuttal, but preferably longer) in the camera ready version.

Strengths: This paper provided the first consistency result for this problem (estimating nonparamatric mixture models from grouped observations).

Weaknesses: The theoretical results are a little weaker than I would like to see, in particular: - Lacking finite-sample guarantees: the convergence of the estimator only holds in the infinity-sample limit (n -> \infty). The finite-sample convergence rate is not established. - Computationally inefficient estimator: the optimization problem (7) is non-convex, hence cannot be solved in poly-time.

Correctness: I have verified that the mathematical claims are correct. The methodology of the experiments look correct to me, but I haven't checked the details of implementation.

Clarity: The paper is mostly well-written. I have made a few minor suggestions for improvement, see detailed comments below.

Relation to Prior Work: The discussion on the prior work is mostly clear, but I hope this part can be expanded a little (see the detailed comments on presentation below).

Reproducibility: Yes

Additional Feedback: My evaluation to this paper is on the borderline - that being said, I am happy to change my score if the authors are able address the following questions: - What's the main obstacle to stronger theoretical guarantees? In particular, (1) Statistically: Is it possible to achieve a finite-sample convergence rate? If the answer is 'no', what makes it difficult? The classical works on KDE have showed that we can achieve a rate of convergence for density estimation under standard assumptions like the Holder smoothness of the density, can we get some results with a similar flavor here? (2) Computationally: Is it possible to achieve an efficient algorithm for this problem? If the answer is 'no', what makes it difficult? The less significant ones: - In the synthetic experiment, the results presented here showed that NDIGO (the algorithm introduced by this work) outperforms other methods on the training sample, but not for out-of-sample setting (where NPMIX performs the best). Can you comment on the possible reasons for this discrepancy? I would like to mention a few possible improvements to the paper below: Presentation: - I think this paper will benefit from a discussion section, especially on why this problem is important (the authors provided two examples, but I hope this part can be more detailed in the future version) and what are the main technical barriers (especially, comparing to non-grouped version of the problem, and density estimation). - In the current version, it was not immediately clear to me why (6) and (7) are equivalent, and why the first term is an empirical quantity. I was able to figure it out on my own, but I hope the authors can provide a brief comment on this (e.g. provide a pointer to supplementary S2.1). Experiments: It would be nice to see a little more experiments on synthetic data (to get better visualization of the clustering induced by the algorithm). For example, the ones used in [10]. However, I fully understand that the focus of this paper is mostly theoretical, so this is just a minor suggestion for the authors.


Review 2

Summary and Contributions: The paper presents a novel algorithm based on weighted kernel density estimators to estimates arbitrary identifiable mixture model from grouped observations. The experimental results show that the approach outperforms baseline methods, especially when mixture components overlap significantly.

Strengths: Authors propose a weighted kernel density estimator for learning mixture models with theoretical guarantees.

Weaknesses: N/A

Correctness: Claims in three theorems look correct without further checking.

Clarity: The paper is easy to follow.

Relation to Prior Work: Authors do not clearly discuss the differences between the proposed work and previous contributions.

Reproducibility: Yes

Additional Feedback:


Review 3

Summary and Contributions: An important limitation of semi-supervised learning is that unlabeled data is not that useful unless certain conditions are satisfied. A strong condition is identifiability of the mixture distribution according to which the unlabeled samples are distributed. If the mixture is identifiable, one could conceptually estimate the components and class probabilities using unlabeled data with an appropriate estimator and use labeled data to label the components of the mixture. Identifiability from iid samples is a strong requirement, traditionally limited to certain - but not all - families of parametric distributions. Recent results show that if one can obtain groups of samples from each of the underlying distribution, then the mixture becomes identifiable under broad conditions. This paper proposes consistent estimators of the mixture distribution using grouped observations, proves consistency results including results for convergence of estimators of the mixture component and corresponding mixing parameters, describes optimization procedures for the estimation, and shows results on a few datasets.

Strengths: The paper appears to be correct, although I did not attempt to derive the results from the hints in the main paper - I just read the proofs. The work appears to be theoretically sound. The experimental results - which actually relate to the optimization procedure rather than to the general asymptotic result 0 are promising and show that the method outperforms (unsurprisingly) clustering methods when identifying overlapping distributions and actually outperforms a close competitor that also attempts to identify mixture components using grouped data. Results on text date (Russian troll Twitter dataset) are promising although it is not obvious to this reviewer that the paired samples are indeed from the same distribution even if they come from the same tweet (people tend to be scattered on twitter). The optimization method also appears to scale well

Weaknesses: The main limitation of the work is that it applies to cases where grouped samples are available,which is not an incredibly common occurrence.

Correctness: The paper appears to be correct. I strongly believe that the main contribution of the paper is of theoretical nature and I am not that concerned with the experimental results. The methodology of the experiments appears to be correct

Clarity: The paper reads extremely well, notation is clear and well chosen, one could read the paper in a single pass.

Relation to Prior Work: yes, the main results are the estimator consistency theorems and the optimization procedure.

Reproducibility: Yes

Additional Feedback: I did not proofread the paper, but line 77 should read "but they do not" rather than "but they does not"


Review 4

Summary and Contributions: This paper proposes a kernel-based estimator of a nonparametric mixture model in which the user has access to paired observations of the true model. The authors prove that the proposed estimator is consistent, propose a practical algorithm for computing it, apply it to simulated data, and finally consider two interesting applications to nuclear source detection and Twitter data (NLP). #### POST-REBUTTAL I appreciate the authors' response to my questions and addressing many of my minor concerns. After discussion, I have decided to increase my score to a 7, since the authors have clarified some of the fundamental road blocks to some of the theoretical issues regarding time/sample complexity. I appreciate the authors' willingness to discuss applications more seriously. One minor point: In asking "which of the algorithms compared make use of the paired observations, and which do not", my point was that the paper only mentions NPMIX in this regard, but none of the other algorithms. It would be helpful for readers if this was clearly spelled out for every algorithm tested in one place.

Strengths: * The paper is well-written and the approach is quite natural * The consistency results are novel, and the theoretical results are presented coherently with the appropriate amount of detail * The experiments are reasonably complete * The biggest strength in my opinion are the two applications, which help to motivate the model, but I would like to see more detail on these applications (see below)

Weaknesses: * The identifiability of the model has been proved previously in [3], so the main contribution is an estimator * This estimator is based on nonconvex program, and it is not clear that this can be solved to optimality using the author's approach * The results are asymptotic, and no there is no discussion of finite sample complexity (esp. wrt to other methods in the literature) * The applications are a significant strong point, however, they are not given enough attention: More detail in describing how the applications maps to this model would be useful, along with more details in the experiments as well (there is ample room to include more text, e.g. by removing Section 3 and moving proof sketches to the supplement) Given the limitations of the theoretical results mentioned above, much of the merit of this paper rests on its practical significance. On my first reading, I did not pay enough attention to the applications since they have been glossed over by the authors. I was thus concerned about the motivation behind the model itself. But after thinking about this, I am convinced the applications are sufficient to motivate the model, however, I do feel much more attention needs to paid attention to this aspect. Furthermore, in addition to applications, I would appreciate more detail regarding computational aspects. How practical is this approach on large datasets? What guarantees can the authors give regarding the optimization scheme? What tradeoffs and limitations exist in practice, esp. compared to related algorithms? It would be interesting to illustrate (or at least describe) some examples from the proposed method fails, but existing algorithms succeed.

Correctness: I found no major issues with the claims in this paper, but there are some untrue statements that should be corrected (see Q8).

Clarity: Overall, the paper is well-written. I have made some minor suggestions in Q8 below regarding presentation.

Relation to Prior Work: The related work section looks reasonably complete and well-referenced.

Reproducibility: Yes

Additional Feedback: Major comments * L134: The assumption that M is known should be mentioned up front and discussed early on (somewhere in Section 1 as well as beginning of Section 4, right now it is hidden at the end of Section 4) * Limitations of this assumptions should also be discussed in detail * Can this be relaxed? * Section 7: The authors claim an approach for "solving (6)", which is a nonconvex problem, and I could not find a proof of the solvability of this program * What guarantees do the authors have? * Will this approach converge to a stationary point, or a global minimum? There is some discussion at L218-219, but more discussion is needed. * In the experiments, which of the algorithms compared make use of the paired observations, and which do not? It should be mentioned that not all algorithms are on equal footing in this regard. Minor comments * When a statistical model is identifiable, under very weak conditions there is guaranteed to be a consistent estimator (e.g. Le Cam and Schwartz, "A necessary and sufficient condition for the existence of consistent estimates", or the book by Ibragimov and Hasminskii) -- what this work proposes is a *practical* estimator, which is much more useful * L18-20: It would be helpful for the reader to point out some applications for this model here (e.g. authors can simply refer to the two good applications they consider later in the paper) * L63: Authors should mention the drawbacks of clustering using mixture models, e.g. identifiability, sample complexity, and the fact that MMs can lead to undesirable or misspecified clusterings * L77-78: The claim about the paper [10] seems incorrect; my understanding based on their Theorem 4.3 is that they do prove recovery of the underlying components * L138: What does it mean for p to be identifiable? Presumably the authors mean the underlying parameters are identifiable. This should be clarified (or appropriately defined somewhere) * L187: This sentence could be phrased better -- Theorems 1 and 2 imply consistency in estimating q, which is of course implies identifiability of q, but this does not necessarily imply identifiability of p: It would read better to clarify this distinction more carefully (e.g. it took me a second to parse what was meant here) * Obviously a matter of personal taste, but I found "NoMM" to be an odd choice of acronym (even confusing in some places); either NPMM or NMM seems more natural Typos * L132: "in the supplemental." * L233: "concision"

[Author Response · NeurIPS 2020]

We thank all reviewers for their comments and we will make every effort to address all comments in the revision.

**R1 and R5: Rates/finite sample bounds.** One of main difficulties in finding a rate of convergence of our estimator is precisely characterizing how convergence of the mixture components (l. 192) depends on the convergence of the mixture distribution (l. 191) in Theorem 3. Our algorithm can be seen as approximating a distribution with a low-rank symmetric tensor. While matrix convergence implies convergence of spectral decompositions and is well characterized, general tensors suffer from a "lack of closeness[1]." E.g. it is possible to construct a sequence of symmetric rank 3 tensors that converges to a symmetric rank 4 tensor[1]. This sort of phenomenon makes it difficult to characterize convergence, and this is only exacerbated since we are working in tensor spaces of functions rather than Euclidean vectors. Interestingly the full version of Theorem 3 (Supp.) implies that for general product spaces "lack of closeness" is solved by assuming densities of the form in (l. 117) identifiable. We will look into this convergence in future work.

**R1 and R5: Nonconvex Optimization** The reviewers have correctly assessed that the suggested algorithm is not guaranteed to solve (6) globally. We note that the class of nonconvex optimization problems known to be solvable by descent methods is extremely limited, e.g., phase retrieval, PCA, matrix completion. The objective function we consider is a polynomial of degree six, while the constraint set is convex. While general polynomial optimization is NP-hard[2], we expect and observe that the proposed optimization problem is no more difficult than other common nonconvex optimization problems in machine learning, e.g., training neural networks. Projected SGD is widely used in practice, and there exists some convergence analysis in the literature [3]. Specifically, convergence to a stationary point can be shown under Lipschitz-type assumptions. In the revision, we will add some discussion of the technical details to the main text, making it clear that we do not claim the proposed algorithm finds a global minimum.

**R1, R3, and R5: Additional Discussion** It was suggested that the work would benefit from more motivation and discussion of applications. As the reviewers have pointed out, the grouped sample setting is not commonly considered in the literature. One exception is clustering with must-link constraints. There is quite a bit of work in this area with applications in interactive visual clustering and computer vision[4]. Additionally our method can be seen as a continuous version of multinomial mixture modelling, which is used in psychometrics where measurements over time are collected for a group of, for example, bipolar disorder patients and used to identify subgroups within that population whose condition is only evident with repeated temporal measurements [5]. We will add more discussion of applications and assumptions to the revised paper by folding section 3 into section 4 and removing one or more proof sketches.

**R1: "NDIGO (the algorithm introduced by this work) outperforms other methods on the training sample, but not for out-of-sample setting (where NPMIX performs the best)."** This is a typo. NDIGO performs best (see supp.).

**R1: "In the current version, it was not immediately clear to me why (6) and (7) are equivalent."** In the main text, we will add more detailed pointers to appropriate sections in the supplement.

**R1: "It would be nice to see a little more experiments on synthetic data."** Before submission, we removed some synthetic data experiments (e.g., variations on the moons dataset, mixtures of mixtures of Gaussians) to keep the paper within eight pages, but we will add these results to the supplemental in the revision.

**R5: "How practical is this approach on large datasets?"** The coreset approach appears to scale very well to large datasets. For example, the Twitter experiment we have $2n > 3 \cdot 10^6$.

**R5: "What tradeoffs and limitations exist in practice, esp. compared to related algorithms?"** In the optimization problem we must tradeoff between computational complexity and clustering performance by adjusting the coreset size. Another limitation of our approach is when mixture components are technically mutually irreducible but their supports only differ on a set of very small measure, but any algorithm may fail in this case. It is hard to point to an algorithm that will work when our's does not since other approaches rely on notions of cluster separability. If the data is well fit by some parametric mixture or if components are well separated it could be better to use existing techniques.

**R5: "The assumption that M is known should be mentioned up front and discussed early on"** We will be sure to emphasize this point in the revision. Although we did not try it, our spectral initialization should offer a heuristic for selecting M by looking for the knee in the eigenvalue curve. AIC/BIC/MDL should also be applicable.

**R5: "In the experiments, which of the algorithms compared make use of the paired observations, and which do not?"** On line 257 we specify that NPMIX does not utilize pair information, but we will reword things to clarify this.

## Footnotes

[1] See "Symmetric tensors and symmetric tensor rank" by Comon et al.

[2] See "Complete solutions and extremality criteria to polynomial optimization problems" by Gao

[3] See "Convergence of a multi-agent projected stochastic gradient algorithm for non-convex optimization" by Bianchi and Jakubowicz, as well as "Zeroth-order Stochastic Projected Gradient Descent For Nonconvex Optimization" by Liu et al.

[4] See "Constrained Clustering: Advances in Algorithms, Theory, and Applications" by Basu et al.

[5] For example "A mixed-binomial model for Likert-type personality measures" Allik 2014 or "Cognitive psychometrics: Using multinomial processing tree models as measurement tools" Batchelder 2010


[Meta-Review · NeurIPS 2020]

Although the reviewers have raised legitimate concerns regarding the theory presented in this paper as well as the motivation and applicability of the grouped observation model, nonetheless a majority agreed the underlying ideas are relevant and the general problem studied is significant. In the camera ready, please pay close attention to the corrections and suggestions pointed out by the reviewers (particularly R1, R5) and expand the discussion of applications, as agreed in the author response.